# Fair and Welfare-Efficient
# Constrained Multi-matchings under Uncertainty

**Elita Lobo**,* **Justin Payan**,* **Cyrus Cousins, and Yair Zick**
University of Massachusetts Amherst
{elobo, jpayan, cbcousins, yzick}@umass.edu

## Abstract

We study fair allocation of constrained resources, where a market designer optimizes overall welfare while maintaining group fairness. In many large-scale settings, utilities are not known in advance, but are instead observed after realizing the allocation. We therefore *estimate* agent utilities using machine learning. Optimizing over estimates requires trading-off between mean utilities and their predictive variances. We discuss these trade-offs under two paradigms for preference modeling – in the *stochastic optimization* regime, the market designer has access to a probability distribution over utilities, and in the *robust optimization* regime they have access to an *uncertainty set* containing the true utilities with high probability. We discuss utilitarian and egalitarian welfare objectives, and we explore how to optimize for them under stochastic and robust paradigms. We demonstrate the efficacy of our approaches on three publicly available conference reviewer assignment datasets. The approaches presented enable scalable constrained resource allocation under uncertainty for many combinations of objectives and preference models.

## 1 Introduction

Constrained resource allocation without money underpins many important systems, including reviewer assignment for peer review (our primary example throughout the paper) [4, 16, 31, 45, 54], assigning resources to homeless populations [5, 34, 49], distributing emergency response resources [51, 56, 57], and more [1, 44, 53]. In these settings, we assign *resources* to *agents*. Agents and resources are *constrained*; each agent has bounds on the minimum or maximum number of items they receive from different categories, and each item has required minimums and limited total capacity. Each agent has a valuation for every item, and we optimize a welfare function of the agent-item valuations. In the case of reviewer assignment, the reviewer-paper valuations measure the alignment between reviewers and papers, papers must receive a certain number of reviews from unique reviewers, reviewers have upper limits on the number of papers they can review, and conflicts of interest prevent some reviewers from being assigned to certain papers.

A crucial factor in all of the above settings is the presence of *uncertainty*. Uncertainty often stems from the fact that agents' valuations for resources depend on future outcomes. In reviewer assignment, a reviewer-paper pair's match quality is observed only after the reviewer submits his or her review. Uncertainty may also stem from our limited ability to collect data; for example, in deciding where to target lead pipe mitigation projects based on number of school-aged children per neighborhood, we may have access to imperfect school enrollment records, allowing only an approximate model of the impacts of mitigation on children in each neighborhood [53]. We adopt two possible stances towards uncertainty, depending on the information available. When we have access to *a*

---

*Authors contributed equally.

38th Conference on Neural Information Processing Systems (NeurIPS 2024).

*probability distribution* over preferences, we optimize the conditional expectation of the distribution at percentiles of interest [33, 50]. When we have access to *a set* of possible preferences, we adopt the *robust* approach, which is related to the minimax regret objective used in solving robust assignment problems [3, 10, 11, 32]. Uncertainty-aware optimization approaches can often result in significantly different allocations from the default of optimizing for welfare over a central estimate (see Example 2.1 for an intuitive explanation for this phenomenon).

Typically, we maximize the sum of agent utilities. However, in many of these settings, we are also concerned with *fairness* to individuals or groups of agents. Groups of agents may represent subject areas of papers in reviewer assignment, demographic groups in poverty alleviation campaigns, or regional groupings of computational resources in bandwidth allocation. Fairness to these groups may be legally required in some cases; in others it is an ethical choice by the decision maker. Although groups are often first-class objects worthy of receiving fair treatment, group fairness is often the smallest granularity of fairness achievable under uncertainty – in a large dataset uncertainty will always cause some individuals to have vanishing welfare, but group welfare can still be upheld. Although there is much literature on combinatorial optimization under uncertainty [3, 10, 11, 32, 33], to our knowledge it has not addressed the intersection of fairness and uncertainty in the constrained multi-matching problem.

## 1.1 Our Contributions

We study the broad problem of fair and efficient constrained multi-matchings under uncertainty about agents' valuations. We optimize for welfare while simultaneously accounting for the uncertainty inherent in real-world resource allocation problems. Specifically, we develop methods to efficiently optimize the utilitarian and egalitarian welfare objectives using the robust approach [7, 8, 26] and the CVaR approach [50]. Our results are summarized in Table 1.

For robust optimization, we construct an uncertainty set that contains the true preferences with high probability (Section 3). This model is appropriate when building a predictor with statistical error bounds, but without making any assumptions on the full probability distribution over valuations. For utilitarian and egalitarian welfare functions, we robustly maximize welfare over such uncertainty sets. When the uncertainty sets are linear, we can efficiently compute the exact optimal allocations for both utilitarian and egalitarian welfare in polynomial time (Corollaries 3.2 and 3.6). Under a single ellipsoidal uncertainty set, we can apply an *iterated quadratic programming* approach (Corollaries 3.3 and 3.7), while a projected subgradient ascent approach is needed when uncertainty sets consist of multiple ellipsoids (Propositions 3.1 and 3.5). Under general monotonic, concave welfare functions and arbitrary convex uncertainty sets, we apply the relatively expensive adversarial projected subgradient ascent algorithm of Cousins *et al.* [16].

When the market designer can construct a full probability distribution over preferences or sample from such a distribution, we consider *stochastic optimization* using the concept of *Conditional Value at Risk*, or CVaR [50]. This approach, laid out in Section 4, selects an allocation that maximizes the conditional expectation of welfare over the left tail of the welfare distribution. We often approximate CVaR objectives using sampling and then solve the resulting linear program or LP (as in Propositions 4.1 and H.3). However, in the case of utilitarian welfare and Gaussian-distributed valuations, we present a simple reformulation of the CVaR objective (Proposition 4.3). Optimizing CVaR for general monotonic, concave welfare functions can require solving arbitrary concave optimization problems, even after sampling.

We also compare these optimization approaches empirically in Section 5 on reviewer assignment data from AAMAS 2015, 2016, and 2021.

## 1.2 Related work

We discuss the history of prior work on robust and CVaR optimization in Appendix A.

Some existing work applies stochastic or robust optimization to fair division problems. A line of work studies the minimax regret objective in combinatorial optimization problems, such as constrained resource allocation [3, 10, 11, 32]. This work does not explicitly consider multi-matching problems like those considered here, nor does it address the robust egalitarian welfare problem. Pujol *et al.* [48] study fair division problems with parameters noised for differential privacy, showing

Table 1: Summary of optimization algorithms for efficiently computing utilitarian and egalitarian welfare under different robustness concepts. Green highlights indicate problems which require solving a single linear program (low difficulty). Yellow highlights indicate solving a small number of linear or quadratic programs (medium difficulty). Red highlights indicate problems which require solving numerous quadratic programs or arbitrary concave programs.

| | Robustness Concept | | | | |
| --- | --- | --- | --- | --- | --- |
| | Robust | | | CVaR | |
| | **Linear** | **One Ellipsoid** | **$\ell$ Ellipsoids** | **Any (approx.)** | **Gaussian** |
| **Utilitarian** | LP Reduction (Coro. 3.2) | Iterated QP (Coro. 3.3) | Projected SGA (Prop. 3.1) | Sampling + LP (Prop. 4.1) | Projected GA (Prop. 4.3) |
| **Egalitarian** | LP Reduction (Coro. 3.6) | Iterated QP (Coro. 3.7) | Projected SGA (Prop. 3.5) | Sampling + LP (Prop. H.3) | |
| **Monotone, Concave** | Adversarial Projected SGA (Sec. 3.3) | | | Sampling + Concave Program (Sec. 4) | |

that the noise can cause unfair allocations; they propose a Monte Carlo approach to mitigate unfairness with high probability. Peters *et al.* [46] study envy-free rent division under probabilistic uncertainty. A central mechanism divides rooms and sets room prices for items to minimize envy. We study a setting without money, both utilitarian and egalitarian objectives, and robust optimization in addition to stochastic optimization.

Cousins *et al.* [16] study robust optimization under the utilitarian objective. They propose an adversarial projected subgradient ascent method, which requires solving two quadratic programs (one for the adversary and one for the projection) at each iteration for a large number of iterations. Our empirical analysis in Section 5 demonstrates the inefficiency of this method. Fair machine learning algorithms [17, 22, 23, 43, 59] often employ similar adversarial optimization techniques over an uncertainty set in a machine learning context. Other fair allocation research has studied the case where agent demand or item availability are uncertain but preferences are known [2, 14, 21, 27]. In our case, demand and availability are known but preferences are not. Devic *et al.* [20] consider fair two-sided matching where the fairness constraint is defined with respect to unknown parameters; we assume knowledge of the parameters that define the fairness constraint (i.e., group identities).

## 2 Fair Resource Allocation under Uncertainty

We first introduce the problem of resource allocation without uncertainty, then lay out the two approaches we take to deal with the introduction of uncertainty. Our results are summarized in Table 1.

### 2.1 Fair Resource Allocation

We have a set of $n$ agents $N = \{a_1, \ldots a_n\}$, and $m$ item types $I = \{i_1, \ldots i_m\}$. Agents are partitioned into $g$ groups $\mathcal{G} = \{G_1, \ldots G_g\}$, with each $G \subseteq N$ and each agent $i$ belonging to exactly one group.

For any $n \times m$ matrix $\boldsymbol{X}$ we use the same lower-case bold letter, i.e., $\boldsymbol{x}$ to denote the vector representing the vectorized form of the matrix $\boldsymbol{X}$, in row-major order. For any group of agents $G$, we use $\mathbf{x}_G \in \mathbb{R}^{|G|m}$ to denote the vector restricted to the agents in $G$. Given vectors $\mathbf{x}, \mathbf{y} \subseteq \mathbb{R}^{nm}$ and real number $c \in \mathbb{R}$, let $\mathbf{x} \succeq c$ denote $\mathrm{x}_j \geq c$ for all $j$, and let $\mathbf{x} \succeq \mathbf{y}$ denote that $\mathbf{x} - \mathbf{y} \succeq 0$. The $\preceq$ operator is defined analogously.

We assume a valuation matrix $\boldsymbol{V}^* \in \mathbb{R}_{0+}^{n \times m}$, where $\boldsymbol{V}_{a,i}^*$ encodes the true value of assigning item type $i$ to agent $a$. The values of $\boldsymbol{V}^*$ are typically unknown; we discuss our approaches to handle this problem in Section 2.2. We use tildes to denote random variables; for example, $\tilde{x} \sim \mathcal{D}_{\tilde{x}}$ denotes that the random variable $\tilde{x}$ follows the distribution $\mathcal{D}_{\tilde{x}}$.

Given some set of feasible assignments $\mathcal{A} \subseteq \mathbb{N}^{n \times m}$, we aim to find assignments $\boldsymbol{A} \in \mathcal{A}$ where $\boldsymbol{A}_{a,i}$ indicates the number of items of type $i$ allocated to agent $a$. For each agent $a \in N$, we have

upper and lower bounds on assignments of the form $\underline{\kappa}_a \leq \sum_{i \in I} \boldsymbol{A}_{a,i} \leq \bar{\kappa}_a$. For each item $i$, we have lower and upper bounds on the total assignment of that item; $\underline{\psi}_i \leq \sum_{a \in N} \boldsymbol{A}_{a,i} \leq \bar{\psi}_i$. Finally, we have pairwise limits $\boldsymbol{C}_{a,i}$ for each agent $a$ and item type $i$, requiring that $\boldsymbol{A}_{a,i} \leq \boldsymbol{C}_{a,i}$. It is always the case that the constraints define a finite set such that $|\mathcal{A}| \in \mathbb{N}$. In the example of reviewer assignment, these constraints reflect the review requirements per paper, load bounds for reviewers, conflicts of interest, and the constraint that no reviewer is assigned twice to any given paper.

Let $\mathbf{u} : \mathcal{A} \times \mathbb{R}_{0+}^{n \times m} \to \mathbb{R}^g$ be an affine function mapping from allocations to utilities for each group. $u_G(\mathbf{a}, \mathbf{v})$ denotes the utility of the group $G$ under allocation $\mathbf{a} \in \mathcal{A}$ (recall $\mathbf{a}$ is the vectorized version of the assignment $\boldsymbol{A}$). We write $\mathbf{u}$ instead of $\mathbf{u}(\mathbf{a}, \mathbf{v})$ when $\mathbf{a}$ and $\mathbf{v}$ are clear from context. We assume that $\mathbf{u}$ is *additive* and normalized by group size, so $u_G = \frac{\mathbf{a}_G^\mathsf{T} \mathbf{v}_G}{|G|}$. We define a *welfare function* $\mathrm{W} : \mathbb{R}^g \to \mathbb{R}$, where $\mathrm{W}(\mathbf{u}(\mathbf{a}, \mathbf{v}))$ denotes the overall welfare of allocation $\mathbf{a}$. The *weighted utilitarian social welfare* function is defined as $\mathbf{w} \cdot \mathbf{u}$, where $\mathbf{w} \in \mathbb{R}_{0+}^g$ denotes the weights on groups in $\mathcal{G}$. When $w_G = |G|$ for all $G$, we call this function simply "utilitarian welfare" or "USW." The *group egalitarian social welfare* function (also "group egalitarian welfare" or "GESW") is defined as $\min_{G \in \mathcal{G}} u_G$. We do not consider individual egalitarian welfare in this work; under robust and stochastic optimization the egalitarian welfare is zero when the number of items is proportional to the number of agents and uncertainty is non-trivial.

## 2.2 Optimizing Allocations under Uncertainty

We consider two main approaches to dealing with uncertainty: the *robust optimization* approach and the *Conditional Value at Risk* approach.

In the robust approach (Section 3), we obtain an *uncertainty set* $\mathcal{V}$ that contains the true agent valuations $\mathbf{v}^*$ with probability at least $1 - \alpha$ for some confidence parameter $\alpha \in [0, 1)$. We then optimize the welfare corresponding to the *worst* valuation matrix in the uncertainty set, i.e., $\max_{\mathbf{a} \in \mathcal{A}} \min_{\mathbf{v} \in \mathcal{V}} \mathrm{W}(\mathbf{u}(\mathbf{a}, \mathbf{v}))$. This approach applies when we do not have access to a full distribution over valuations but have some other way of describing $\mathcal{V}$ [16].

When we have access to a complete distribution of the random variable $\tilde{\mathbf{v}} \in \mathbb{R}^{nm}$, we apply a stochastic approach instead. We compute the welfare distribution and optimize the conditional expectation over an $\alpha$-percentile of the welfare or *Conditional Value at Risk at $\alpha$* ($\mathrm{CVaR}_\alpha$). Suppose that we have a random variable $\tilde{x} \sim \mathcal{D}_{\tilde{x}}$. For any risk level $\alpha \in (0, 0.5)$, $\mathrm{CVaR}_\alpha[\tilde{x}]$ is defined as $\mathbb{E}_{\tilde{x} \sim \mathcal{D}_{\tilde{x}}}[\tilde{x} \mid \tilde{x} \leq \nu_\alpha]$ where $\nu_\alpha$ denotes the $\alpha$-percentile of $\tilde{x}$. This approach is only appropriate when $\mathcal{D}_{\tilde{x}}$ is fully known, or when we can sample from it. We investigate this regime in Section 4, where we will compute and optimize $\mathrm{CVaR}_\alpha[\mathrm{W}(\mathbf{u}(\mathbf{a}, \tilde{\mathbf{v}}))]$ for a random variable $\tilde{\mathbf{v}} \sim \mathcal{D}_{\tilde{\mathbf{v}}}$.

**Example 2.1** (The Importance of Considering Uncertainty). Consider a simple two-agent, two-item instance, where each agent needs to get exactly one item, and either likes (utility 1) or dislikes it (utility 0). Agent preferences are Bernoulli random variables, where $\Pr[\tilde{v}_{1,1} = 1] = 0.8, \Pr[\tilde{v}_{1,2} = 1] = 0.9, \Pr[\tilde{v}_{2,1} = 1] = 0.5$, and $\Pr[\tilde{v}_{2,2} = 1] = 0.8$. If we maximize the expected USW, we would assign $i_1$ to $a_1$ and $i_2$ to $a_2$, for a total expected USW of 1.6. However, consider instead the $\mathrm{CVaR}_{0.3}$ of USW. When we make the expectation-maximizing assignment, then $\Pr[\mathrm{W}(\mathbf{u}) = 0] = 0.04$ and $\Pr[\mathrm{W}(\mathbf{u}) = 1] = 0.32$. However, if we assign $i_2$ to agent $a_1$ and item $i_1$ to agent $a_2$, we have $\Pr[\mathrm{W}(\mathbf{u}) = 0] = 0.05$ and $\Pr[\mathrm{W}(\mathbf{u}) = 1] = 0.5$. This means that the conditional expectation of welfare at the $30^{th}$ percentile is higher if we assign $i_2$ to $a_1$ and $i_1$ to $a_2$ (it is .32 in the first case and .5 in the second case). If we want to retain welfare in the face of uncertainty, we might well choose to maximize this quantity rather than the expectation of the welfare.

## 3 Robust Welfare Optimization

We construct the optimization problems for utilitarian and egalitarian welfare objectives with the robust approach. Many of these optimization problems are concave-convex max-min problems that can be directly solved using an adversarial projected subgradient ascent technique [16]: in each iteration of the algorithm, the inner minimization problem is solved to optimality, followed by a subgradient step on the allocation $\mathbf{a}$, followed by a projection onto the constraint space $\mathcal{A}$. However, this method does not exploit the structure of these problems and is often computationally expensive or intractable, as demonstrated empirically in Section 5. Despite the inherent complexities of these problems, we show that, under specific assumptions, these problems can be reduced to more

manageable forms that are easier to optimize. We then discuss a range of algorithms for efficiently optimizing the simplified problems.

**Scope:** The robust approach detailed in Section 2 assumes the availability of an uncertainty set of the valuation matrix. For the sake of computational tractability, we focus on the class of uncertainty sets defined by linear and ellipsoidal constraints

$$\mathcal{V} = \left\{ \mathbf{v} \in \mathbb{R}^{nm} \mid \forall i \in [1, \ell], \, (\mathbf{v} - \bar{\mathbf{v}}_i)^{\mathsf{T}} \mathbf{S}_i^{-1} (\mathbf{v} - \bar{\mathbf{v}}_i) \leq r_i^2, \boldsymbol{Q}\mathbf{v} \succeq \mathbf{e}, \mathbf{v} \succeq 0 \right\} \,,$$

where the $i^{th}$ ellipsoid is centered at $\bar{\mathbf{v}}_i \in \mathbb{R}_{0+}^{nm}$ with a positive definite covariance matrix $\mathbf{S}_i \in \mathbb{R}^{nm \times nm}$ and radius $r_i \in \mathbb{R}_{0+}$, and $\boldsymbol{Q} \in \mathbb{R}^{k \times nm}$, and $\mathbf{e} \in \mathbb{R}^k$ define additional linear constraints.

This limitation on the structure of uncertainty sets is not too restrictive; it is possible to construct such uncertainty sets for linear regression and logistic regression models using statistical bounds, as shown in Appendix D.

In all of our methods, where obtaining an integer allocation is either not feasible or computationally tractable, we relax the set of feasible integer assignments $\mathcal{A} \subseteq \mathbb{N}^{n \times m}$ to a set of feasible continuous allocations $\mathcal{A} \subseteq \mathbb{R}_{0+}^{n \times m}$. One can obtain integer allocations satisfying all constraints by applying a randomized rounding technique that generalizes the Birkhoff-von Neumann decomposition [13]. The fractional solutions can thus be interpreted as randomized allocations.

## 3.1 Robust Allocation for Utilitarian Welfare

We consider the problem of finding an allocation that optimizes the weighted utilitarian welfare under the worst valuation matrix in the uncertainty set. We formulate the problem as

$$\max_{\mathbf{a} \in \mathcal{A}} \min_{\mathbf{v} \in \mathcal{V}} \mathbf{w} \cdot \mathbf{u}(\mathbf{a}, \mathbf{v}) \,. \tag{1}$$

The objective and constraints of the inner-minimization problem described in (1) are convex. The problem is strictly feasible, which satisfies Slater's condition [12] for strong duality. Therefore, by taking the dual of the inner-minimization problem, we can simplify the problem in (1) into a single equivalent maximization problem. We provide the dual formation in Proposition 3.1.

To simplify the notation in the results that follow, we assume, without loss of generality, that each group $G$ has weight $\mathrm{w}_G = |G|$. In practice, if this assumption does not hold, the weights can be incorporated into the valuations $\mathbf{v}$ with a corresponding adjustment to the parameters of the valuation uncertainty set $\mathcal{V}$.

In the dual, let $\boldsymbol{\beta} \in \mathbb{R}_{0+}^k$ be the dual variable corresponding to the linear constraints $\boldsymbol{Q}\mathbf{v} \succeq \mathbf{e}$, $\boldsymbol{\lambda} \in \mathbb{R}_{0+}^{\ell}$ be the dual variable associated with the ellipsoidal constraints, and $\boldsymbol{\xi} \in \mathbb{R}^{nm}$ be the variable that combines the primal variable $\mathbf{a}$ with the dual variable of the non-negativity constraint on $\mathbf{v}$ for variable elimination. We define a set of feasible $\boldsymbol{\xi}$ as

$$\Lambda \doteq \mathcal{A} - \mathbb{R}_{0+}^{nm} = \left\{ \boldsymbol{\xi} \in \mathbb{R}^{nm} \, \middle| \, \forall a \in N : \sum_{i \in I} \xi_{am+i} \leq \bar{\kappa}_a, \forall i \in I : \sum_{a \in N} \xi_{am+i} \leq \bar{\psi}_i, \boldsymbol{\xi} \preceq \boldsymbol{c} \right\} \,, \tag{2}$$

which is Pareto-dominated by $\mathcal{A}$.

**Proposition 3.1** (Robust Utilitarian Welfare Dual)**.** *The problem in* (1) *is equivalent to solving*

$$\max_{\substack{\boldsymbol{\xi} \in \Lambda \\ \boldsymbol{\lambda} \in \mathbb{R}_{0+}^l, \boldsymbol{\beta} \in \mathbb{R}_{0+}^k}} \mathbf{p}^{\mathsf{T}} \mathbf{T} \mathbf{q} + \boldsymbol{\beta}^{\mathsf{T}} \mathbf{e} - \frac{1}{4} \mathbf{p}^{\mathsf{T}} \mathbf{T} \mathbf{p} + \sum_{i=1}^{\ell} \left( \lambda_i \bar{\mathbf{v}}_i^{\mathsf{T}} \mathbf{S}_i^{-1} \bar{\mathbf{v}}_i - \lambda_i r_i^2 \right) - \mathbf{q}^{\mathsf{T}} \mathbf{T} \mathbf{q} \,, \tag{3}$$

*where* $\mathbf{p} = \boldsymbol{\xi} - \boldsymbol{Q}^{\mathsf{T}} \boldsymbol{\beta}$, $\mathbf{q} = \sum_{i=1}^{\ell} \lambda_i \mathbf{S}_i^{-1} \bar{\mathbf{v}}_i$, *and* $\mathbf{T} = \left( \sum_{i=1}^{\ell} \lambda_i \mathbf{S}_i^{-1} \right)^{-1}$. *Let* $\boldsymbol{\xi}^*$ *be the optimal* $\boldsymbol{\xi}$ *in* (3). *Then the optimal allocation* $\mathbf{a}^*$ *can be derived from* $\boldsymbol{\xi}^*$ *by finding* $\mathbf{a} \in \mathcal{A}$ *such that* $\mathbf{a} \preceq \boldsymbol{\xi}^*$.

Proposition 3.1 shows that the optimal allocation for the problem in Equation (1) can be computed by first solving the concave program in Equation (3) to obtain $\boldsymbol{\xi}^*$ and then deriving the optimal allocation $\mathbf{a}^*$ from $\boldsymbol{\xi}^*$ by solving a system of equations. Notably, the problem in Equation (3) is a single maximization problem with fewer variables and constraints as compared to the max-min problem in (1), making it simpler to solve. We can either solve it using off-the-shelf convex optimization tools, or by applying a projected subgradient ascent approach (without the previously required adversary).

When the valuation uncertainty set is polyhedral, the problem in (3) simplifies further into a linear program (LP) which can be solved efficiently using standard LP solvers like Gurobi [28].

**Corollary 3.2** (Utilitarian Welfare with Polyhedral Uncertainty). *In the case where the uncertainty set $\mathcal{V}$ is defined purely by linear constraints, i.e., $\mathcal{V} = \{\mathbf{v} \in \mathbb{R}^{nm} \mid \mathbf{Q}\mathbf{v} \succeq \mathbf{e}, \mathbf{v} \succeq 0\}$, the optimal allocation $\mathbf{a}^*$ for the problem in (1) can be computed by solving the linear program*

$$\max_{\mathbf{a} \in \mathcal{A}, \boldsymbol{\beta} \in \mathbb{R}_{0+}^{k}} \boldsymbol{\beta}^\mathsf{T} \mathbf{e} \quad s.t. \ \boldsymbol{Q}^\mathsf{T} \boldsymbol{\beta} \preceq \mathbf{a} \ .$$

When the valuation uncertainty set has a single ellipsoidal constraint with a non-negativity constraint, we compute the solution using iterated quadratic programming (Iterated QP).

**Corollary 3.3** (Utilitarian Welfare with Ellipsoidal Uncertainty). *Suppose that the set $\mathcal{V}$ in (1) is defined by a single truncated ellipsoidal constraint, i.e., $\mathcal{V} = \{\mathbf{v} \in \mathbb{R}^{nm} \mid (\mathbf{v} - \bar{\mathbf{v}})^\mathsf{T} \mathbf{S}^{-1} (\mathbf{v} - \bar{\mathbf{v}}) \leq r^2, \mathbf{v} \succeq 0\}$. The problem in (1) is equivalent to solving*

$$\max_{\lambda \in \mathbb{R}_{0+}, \boldsymbol{\xi} \in \Lambda} \boldsymbol{\xi}^\mathsf{T} \bar{\mathbf{v}} - \frac{\boldsymbol{\xi}^\mathsf{T} \mathbf{S} \boldsymbol{\xi}}{4\lambda} - \lambda r^2 \ . \tag{4}$$

*The exact optimal solution $(\lambda^*, \boldsymbol{\xi}^*)$ to Equation (4) can be computed by alternately performing two steps until convergence: first, fixing $\boldsymbol{\xi}$ and optimizing $\lambda$, i.e., $\lambda = \sqrt{\boldsymbol{\xi}^\mathsf{T} \mathbf{S} \boldsymbol{\xi}}/2r$, and second, fixing $\lambda$ and solving a concave quadratic program to optimize $\boldsymbol{\xi}$. The optimal allocation $\mathbf{a}^*$ can be computed from $\boldsymbol{\xi}^*$ as in Proposition 3.1.*

## 3.2 Robust Allocation for Group Egalitarian Welfare

We now consider the problem of maximizing egalitarian welfare under the robust approach. We can represent this problem as

$$\max_{\mathbf{a} \in \mathcal{A}} \min_{\mathbf{v} \in \mathcal{V}} \min_{G \in \mathcal{G}} \mathrm{u}_G(\mathbf{a}, \mathbf{v}) \ . \tag{5}$$

This problem presents inherent challenges due to the non-smoothness of the inner-minimization problem and the joint constraint on the uncertainties of the valuation matrices of different groups. These factors make it difficult to compute the dual and reduce the problem or efficiently solve the problem using the quadratic program technique described in Corollary 3.3, although the generic adversarial subgradient ascent approach of Cousins *et al.* [16] can still be applied. For the remainder of this section, we assume that the uncertainty sets for each group $G \in \mathcal{G}$ are independent of each other. To simplify notation, we assume, without loss of generality, that the valuations $\mathbf{v}$ and the parameters of the valuation uncertainty set $\mathcal{V}$ are scaled to incorporate the factor $\frac{1}{|G|}$ in the representation of the utility of each group $G$ in the corresponding group valuation $\mathbf{v}_G$.

**Assumption 3.4** (Independence of Groups). *The uncertainty set $\mathcal{V}$ is a Cartesian product of individual groups' uncertainty sets, $\mathcal{V} \doteq \bigotimes_{G \in \mathcal{G}} \mathcal{V}_G$ where each group's uncertainty set $\mathcal{V}_G$ is given by*

$$\mathcal{V}_G = \left\{ \mathbf{v}_G \in \mathbb{R}^{|G|m} \mid \forall i \in [1, \ell], (\mathbf{v}_G - \bar{\mathbf{v}}_{G,i})^\mathsf{T} \mathbf{S}_{G,i}^{-1} (\mathbf{v}_G - \bar{\mathbf{v}}_{G,i}) \leq r_{G,i}^2, \boldsymbol{Q}_G \mathbf{v}_G \succeq \mathbf{e}_G, \mathbf{v}_G \succeq 0 \right\} \ .$$

*Here the $i^{th}$ ellipsoid in group $G$'s uncertainty set is centered at $\bar{\mathbf{v}}_{G,i} \in \mathbb{R}_{0+}^{|G|m}$ with positive definite covariance matrix $\mathbf{S}_{G,i} \in \mathbb{R}^{|G|m \times |G|m}$ and radius $r_{G,i} \in \mathbb{R}_{0+}$, and $\boldsymbol{Q}_G \in \mathbb{R}^{k \times |G|m}$, and $\mathbf{e}_G \in \mathbb{R}^k$ define additional linear constraints.*

This assumption is not unreasonable in practical scenarios. For example, conferences often group papers into disjoint tracks or require paper authors to select a single primary subject area. Although papers may have multiple secondary subject areas, the top-level grouping remains independent. Assumption 3.4 allows us to reorder the two minimization problems without compromising generality:

$$\max_{\mathbf{a} \in \mathcal{A}} \min_{G \in \mathcal{G}} \min_{\mathbf{v}_G \in \mathcal{V}_G} \mathbf{a}_G^\mathsf{T} \mathbf{v}_G \ . \tag{6}$$

We take the dual of the inner minimization problem and then reorder the minimization over groups and the maximization over the dual variables to obtain a single, concave max-min problem. This can be solved with projected subgradient ascent in the general case, or with more efficient approaches in special cases. Proposition 3.5 expresses the general form of the result.

**Proposition 3.5** (Robust Group Egalitarian Dual). *The problem in* (5) *is equivalent to solving*

$$
\begin{aligned}
\max_{\substack{\boldsymbol{\xi}\in\Lambda \\ \boldsymbol{\lambda}\in\mathbb{R}_{0+}^{g\times l} \\ \boldsymbol{\beta}\in\mathbb{R}_{0+}^{g\times k}}} \min_{G\in\mathcal{G}} \; & \boldsymbol{\beta}_G^{\mathsf{T}}\mathbf{e}_G + \mathbf{p}_G^{\mathsf{T}}\mathbf{T}_G\mathbf{q}_G - \frac{1}{4}\mathbf{p}_G^{\mathsf{T}}\mathbf{T}_G\mathbf{p}_G + \\
& \sum_{i=1}^{\ell}\left(\lambda_{G,i}\bar{\mathbf{v}}_{G,i}^{\mathsf{T}}\mathbf{T}_G\bar{\mathbf{v}}_{G,i} - \lambda_{G,i}r_{G,i}^2\right) - \mathbf{q}_G^{\mathsf{T}}\mathbf{T}_G\mathbf{q}_G \;,
\end{aligned}
\tag{7}
$$

*where for each group* $G \in \mathcal{G}$, $\mathbf{p}_G = \boldsymbol{\xi}_G - \boldsymbol{Q}_G^{\mathsf{T}}\boldsymbol{\beta}_G$, $\mathbf{q}_G = \sum_{i=1}^{\ell}\lambda_{G,i}\mathbf{S}_{G,i}^{-1}\bar{\mathbf{v}}_{G,i}$, $\mathbf{T}_G = \left(\sum_{i=1}^{\ell}, \lambda_{G,i}\mathbf{S}_{G,i}^{-1}\right)^{-1}$, *and* $\Lambda$ *is defined as in Equation* (2). *The optimal allocation* $\mathbf{a}^*$ *can be computed from* $\boldsymbol{\xi}^*$ *as in Proposition 3.1.*

The dual variables $\boldsymbol{\lambda}_G, \boldsymbol{\beta}_G, \boldsymbol{\zeta}_G$ and $\boldsymbol{\xi}_G$ for each group $G$ are interpreted as in Proposition 3.1. The optimization problem in (7) is concave with respect to the dual variables $\boldsymbol{\lambda}, \boldsymbol{\beta}$ and $\boldsymbol{\xi}$. We can solve it using an off-the-shelf convex programming library or by applying projected subgradient ascent.

Under polyhedral uncertainty sets, Equation (7) simplifies to a linear program. This is akin to what we observe in the robust utilitarian case (Corollary 3.2).

**Corollary 3.6** (Group Egalitarian Welfare with Polyhedral Uncertainty). *In the case where the uncertainty set* $\mathcal{V}$ *is defined only by linear constraints, i.e.,* $\mathcal{V} = \{\mathbf{v}\in\mathbb{R}^{nm} \mid \forall G\in\mathcal{G} : \boldsymbol{Q}_G\mathbf{v}_G \succeq \mathbf{e}_G, \mathbf{v}\succeq 0\}$, *the max-min-min problem in* (5) *transforms into a linear program.*

When the valuation uncertainty set is defined by a single ellipsoidal constraint per group, we can employ the iterated quadratic programming (Iterated QP) approach used in Corollary 3.3, alternately fixing $\boldsymbol{\lambda}$ and optimizing the rest of the dual variables $(\boldsymbol{\beta}, \boldsymbol{\xi})$ until convergence.

**Corollary 3.7** (Group Egalitarian Welfare with Ellipsoidal Uncertainty). *Suppose that the set* $\mathcal{V}$ *in* (5) *is defined by a single truncated ellipsoidal constraint per group i.e.,* $\mathcal{V} = \{\mathbf{v}\in\mathbb{R}^{nm} \mid \forall G\in\mathcal{G} : (\mathbf{v}_G - \bar{\mathbf{v}}_G)\mathbf{S}_G^{-1}(\mathbf{v}_G - \bar{\mathbf{v}}_G) \leq r_G^2, \mathbf{v}\succeq 0\}$. *Then the problem in* (5) *is equivalent to solving*

$$
\max_{\substack{\boldsymbol{\lambda}\in\mathbb{R}_{0+}^g \\ \boldsymbol{\xi}\in\Lambda}} \min_{G\in\mathcal{G}} \boldsymbol{\xi}_G^{\mathsf{T}}\bar{\mathbf{v}}_G - \frac{\boldsymbol{\xi}_G^{\mathsf{T}}\mathbf{S}_G\boldsymbol{\xi}_G}{4\lambda_G} - \lambda_G r_G^2 \;.
$$

*The exact optimal solution* $(\lambda^*, \boldsymbol{\xi}^*)$ *to Equation* (4) *can be computed by alternately performing two steps until convergence: first, fixing* $\boldsymbol{\xi}$ *and optimizing* $\boldsymbol{\lambda}$, *i.e.,* $\forall G\in\mathcal{G}$, $\lambda_G = \sqrt{\boldsymbol{\xi}_G^{\mathsf{T}}\mathbf{S}_G\boldsymbol{\xi}_G}/2r_G$, *and second, fixing* $\boldsymbol{\lambda}$ *and solving a concave quadratic program to optimize* $\boldsymbol{\xi}$. *The optimal allocation* $\mathbf{a}^*$ *can be computed from* $\boldsymbol{\xi}^*$ *as in Proposition 3.1.*

### 3.3 Robust Allocation for Monotonic Welfare Functions

We now extend our findings to a broader class of monotonic welfare functions. Specifically, we show that when optimizing a monotonic welfare objective under Assumption 3.4, we can decompose the problem into sub-problems such that we independently determine the worst valuation in the uncertainty set of each group, while jointly optimizing the allocation over all groups.

**Proposition 3.8** (Decomposition for Monotonic Welfare Functions). *Consider an optimization problem of the form*

$$
\max_{\mathbf{a}\in\mathcal{A}} \min_{\mathbf{v}\in\mathcal{V}} \; \mathrm{W}_{\mathrm{M}}(\mathbf{u}(\mathbf{a}, \mathbf{v})) \;,
\tag{8}
$$

*where the welfare function* $\mathrm{W}_{\mathrm{M}}$ *is monotonic in the utility of groups. If Assumption 3.4 holds, then* (8) *simplifies to*

$$
\max_{\mathbf{a}\in\mathcal{A}} \mathrm{W}_{\mathrm{M}}\left(\min_{\mathbf{v}_{G_1}\in\mathcal{V}_{G_1}} \mathrm{u}_{G_1}(\mathbf{a}_{G_1}, \mathbf{v}_{G_1}), \min_{\mathbf{v}_{G_2}\in\mathcal{V}_{G_2}} \mathrm{u}_{G_2}(\mathbf{a}_{G_2}, \mathbf{v}_{G_2}), \ldots, \min_{\mathbf{v}_{G_g}\in\mathcal{V}_{G_g}} \mathrm{u}_{G_g}(\mathbf{a}_{G_g}, \mathbf{v}_{G_g})\right) \;.
$$

Proposition 3.8 helps us derive simplified versions of Equation (8), when Assumption 3.4 holds. The egalitarian problem in (5) is an instance of the class of optimization problems described in (8), hence Proposition 3.8 holds under Assumption 3.4 and allows us to derive a single maximization problem (Proposition 3.5). If the allocation and valuation uncertainty sets are convex and compact, the problem in (8) can be solved using constrained convex-concave minimax optimization algorithms [18, 25, 55], or adversarial projected gradient ascent [16]. These approaches do not depend on Assumption 3.4, although optimization can be simplified if independence does hold.

# 4 Stochastic Welfare Optimization

In this section, we optimize the CVaR of utilitarian and egalitarian welfare. This approach works when the distribution $\mathcal{D}_{\tilde{\mathbf{v}}}$ over the valuation matrix is known or when we can sample from $\mathcal{D}_{\tilde{\mathbf{v}}}$. We demonstrate that when the distribution follows a Gaussian distribution, the CVaR of the utilitarian welfare has a simple representation that can be optimized without sample approximation using a projected gradient ascent method. In all other cases, we can approximately optimize CVaR using a sampling-based approach. In particular, when we have monotone, concave welfare functions, we can always approximate the CVaR objective using sampling. However, unlike in Propositions 4.1 and H.3, where the approximated problem becomes linear, with arbitrary monotone, concave welfare functions the problem may require general concave optimization.

## 4.1 CVaR Allocation for Utilitarian Welfare

We wish to find an allocation that maximizes the $\text{CVaR}_\alpha$ of the weighted utilitarian welfare. Let $\tilde{\mathbf{v}}$ represent the random valuation vector. For confidence level $\alpha$, we formulate the problem as

$$\max_{\mathbf{a}\in\mathcal{A}} \text{CVaR}_\alpha\left[\mathbf{w}\cdot\mathbf{u}(\mathbf{a},\tilde{\mathbf{v}})\right] \doteq \max_{\mathbf{a}\in\mathcal{A},b\in\mathbb{R}}\left\{b - \frac{1}{\alpha}\mathop{\mathbb{E}}_{\tilde{\mathbf{v}}\sim\mathcal{D}_{\tilde{\mathbf{v}}}}\left[\left(b-\mathbf{w}\cdot\mathbf{u}(\mathbf{a},\tilde{\mathbf{v}})\right)_+\right]\right\} \ , \tag{9}$$

where $(x)_+ = \max(x, 0)$ represents the positive part of $x$ [50]. Computing the exact expectation in this problem may not be feasible for every distribution $\mathcal{D}_{\tilde{\mathbf{v}}}$. Therefore, we adopt a sampling-based approach. We begin by drawing $h$ i.i.d. samples of the valuation matrix from $\mathcal{D}_{\tilde{\mathbf{v}}}$ represented as $\mathbf{v}^1, \mathbf{v}^2, \mathbf{v}^3, \ldots, \mathbf{v}^h$. We then use these samples to solve the problem described in (9) by solving the linear program outlined in Proposition 4.1.

**Proposition 4.1** (Approximate CVaR of USW). *Given $h$ i.i.d samples of $\tilde{\mathbf{v}}$, i.e., $\mathbf{v}^1, \mathbf{v}^2, \mathbf{v}^3, \ldots, \mathbf{v}^h$ from $\mathcal{D}_{\tilde{\mathbf{v}}}$, the optimal allocation for the problem in* (9) *can be approximately computed by solving*

$$\max_{\mathbf{a}\in\mathcal{A}} \max_{\substack{\boldsymbol{y}\in\mathbb{R}_{0+}^h \\ b\in\mathbb{R}}} \left(b - \frac{1}{\alpha}\sum_{j=1}^h \boldsymbol{y}_j\right) \qquad \forall j\in[1,h]: \ \boldsymbol{y}_j \geq \frac{1}{h}\left(b - \mathbf{w}\cdot\mathbf{u}(\mathbf{a},\mathbf{v}^j)\right) \ . \tag{10}$$

The CVaR estimator used in (10) is a strongly consistent estimator [29]. Therefore, the approximation error of the objective in (10) goes to 0 as $h \to \infty$. In Proposition 4.2, we bound the sample complexity of the problem in (10) when the valuation matrix is sub-Gaussian distributed.

For any allocation $\mathbf{a}$, let $\hat{c}_{h,\alpha}(\mathbf{a})$ represent the empirical estimate of $\text{CVaR}_\alpha[\mathbf{w}\cdot\mathbf{u}(\mathbf{a},\tilde{\mathbf{v}})]$ computed from $h$ samples and $c_\alpha(\mathbf{a})$ represent the corresponding true value. We will use $|\mathcal{A}|$ to denote the number of feasible allocations and $f_{\mathbf{a}} : \mathbb{R} \to \mathbb{R}_{0+}$ to denote the density function of the random welfare $\text{W}(\mathbf{a},\tilde{\mathbf{v}})$. $\nu_\alpha$ denotes the $\alpha$ percentile of $\text{W}(\mathbf{a},\tilde{\mathbf{v}})$.

**Proposition 4.2** (Sample Complexity of Approximate CVaR of USW). *Suppose that $\tilde{\mathbf{v}}$ is a multivariate sub-Gaussian random variable with mean $\bar{\mathbf{v}} \in \mathbb{R}^{nm}$ and covariance proxy $\mathbf{S} \in \mathbb{R}^{nm\times nm}$, i.e., for all vectors $\boldsymbol{z} \in \mathbb{R}^{nm}$ : $\mathbb{E}_{\tilde{\mathbf{v}}\sim\mathcal{D}_{\tilde{\mathbf{v}}}}\left[\exp((\tilde{\mathbf{v}}-\bar{\mathbf{v}})^\intercal\boldsymbol{z})\right] \leq \exp(\boldsymbol{z}^\intercal\mathbf{S}\boldsymbol{z}/2)$, and that, for any risk level $\alpha \in (0, \frac{1}{2})$ and allocation $\mathbf{a} \in \mathcal{A}$, there exists probability density threshold $\eta > 0$ and radius $\gamma > 0$, s.t., $f_{\mathbf{a}}(x) > \eta, \forall x \in [\nu_\alpha - \gamma, \nu_\alpha + \gamma]$. Set $\forall G \in \mathcal{G} : \mathbf{a}'_G = \frac{\mathbf{w}_G\cdot\mathbf{a}_G}{|G|}$. Then, for any confidence parameter $\delta \in (0, 1)$ and error tolerance $\varepsilon > 0$,*

$$\Pr\left[\sup_{\mathbf{a}\in\mathcal{A}}|\hat{c}_{h,\alpha}(\mathbf{a}) - c_\alpha(\mathbf{a})| \leq \varepsilon\right] \geq 1 - \delta \quad for \quad h > \left\lceil\frac{8\max\left(\max_{\mathbf{a}\in\mathcal{A}}\mathbf{a}'^\intercal\mathbf{S}\mathbf{a}', 8\right)\ln\left(\frac{6|\mathcal{A}|}{\delta}\right)}{\min(\varepsilon^2, 16\gamma^2)\alpha^2\min(\eta^2, 1)}\right\rceil \ .$$

Proposition 4.2 follows directly from Theorem 3.1 of L.A. *et al.* [35]. When $\tilde{\mathbf{v}}$ is Gaussian distributed, we can circumvent the sampling approach and instead solve an optimization problem (Proposition 4.3), which depends solely on the mean and covariance of $\tilde{\mathbf{v}}$.

**Proposition 4.3** (CVaR of USW for Gaussian Distributions). *If $\tilde{\mathbf{v}}$ is distributed as a multivariate Gaussian, i.e., $\tilde{\mathbf{v}} \sim \mathcal{N}(\bar{\mathbf{v}}, \mathbf{S})$, then, the optimization problem in* (9) *simplifies to*

$$\max_{\mathbf{a}\in\mathcal{A}} \mathbf{a}'^\intercal\bar{\mathbf{v}} - \frac{\phi(\Phi^{-1}(\alpha))}{\alpha}\sqrt{\mathbf{a}'^\intercal\mathbf{S}\mathbf{a}'} \ , \tag{11}$$

where $\forall G \in \mathcal{G} : \mathbf{a}'_G = \frac{\mathbf{w}_G \cdot \mathbf{a}}{|G|}$, and $\phi$ and $\Phi$ denote the probability density function and the cumulative density function, respectively, of the standard normal distribution $\mathcal{N}(0, 1)$. The problem in (11) is concave and can be solved using the projected gradient ascent method.

## 4.2 CVaR **Allocation for Group Egalitarian Welfare**

For our final objective, we wish to optimize egalitarian welfare under uncertainty using the CVaR approach. We formulate this optimization problem as

$$\max_{\mathbf{a} \in \mathcal{A}} \mathrm{CVaR}_\alpha \left[ \min_{G \in \mathcal{G}} \mathbf{u}_G(\mathbf{a}, \tilde{\mathbf{v}}) \right] \doteq \max_{\mathbf{a} \in \mathcal{A}, b \in \mathbb{R}} \left\{ b - \frac{1}{\alpha} \mathbb{E}_{\tilde{\mathbf{v}} \sim \mathcal{D}_{\tilde{\mathbf{v}}}} \left[ \left( b - \min_{G \in \mathcal{G}} \mathrm{u}_G(\mathbf{a}, \tilde{\mathbf{v}}) \right)_+ \right] \right\} . \qquad (12)$$

To optimize the problem described in the above equation, we solve a linear program similar to the one used to optimize the utilitarian objective CVaR in (9). See Proposition H.3 for more details.

## 5   Experiments

We run experiments on three reviewer assignment datasets. The datasets contain bids from the International Conference on Autonomous Agents and Multiagent Systems (AAMAS) 2015, 2016, and 2021 [41, 42]. We consider the papers as the "agents" and the reviewers as the "items." This is a fairly standard assumption in most recent reviewer assignment approaches, reflecting the primary goal of peer review to assign qualified and interested reviewers to papers [16, 30, 31, 38, 45, 54].

The reviewers issue bids of `yes`, `maybe`, `no`, or `no response`. We run two experiments with this data. In one, we binarize the bids such that `yes` and `maybe` are considered affirmative and `no` is considered negative, while in the other we convert the bids to numerical scores such that `yes` is 1, `maybe` is .5, and `no` is 0.01. Under the binarized model, we fit a logistic matrix factorization model to predict whether the bid is affirmative or negative, and in the continuous model, we fit a Gaussian process matrix factorization model [36]. We derive probability distributions and uncertainty sets from these models. More details on prediction and uncertainty set construction are in Appendix E. These datasets do not contain groups of papers and reviewers, so we create 4 roughly balanced clusters of reviewers and papers for each dataset using the procedure outlined in Appendix F. We define our valid set of assignments $\mathcal{A}$ as follows. For each paper $a \in N$, we set $\underline{\kappa}_a = \bar{\kappa}_a = 3$ for all $a$ in AAMAS 2015, and $\underline{\kappa}_a = \bar{\kappa}_a = 2$ for all $a$ in AAMAS 2016 and 2021. For each reviewer $i$, we set $\underline{\psi}_i = 0$ and $\bar{\psi}_i = 15$ for 2015 and 2016 and 4 for 2021. We optimize and evaluate $\mathrm{CVaR}_{0.01}$; we take $4,000$ samples from the distribution to optimize for CVaR using the sampling-based approach, and we take $10,000$ samples to estimate CVaR for evaluation. We optimize and evaluate robust welfare at the $\alpha = 0.3$ level (there is a $70\%$ chance that the true values lie in the uncertainty set). We constrain the naïve and CVaR approaches to select integer allocations, while the robust approach selects fractional allocations without rounding.

All results are averaged over 5 subsampling runs $20\%$ of each dataset. For each run, we construct 6 allocations, maximizing the naïve central estimate, CVaR and robust statistics for USW and GESW respectively. We evaluate each allocation on each metric. For each run, we normalize each metric by the maximum value achieved for that metric by any allocation. We normalize in this manner to highlight that the allocation targeted for a given objective always returns the highest value on that objective, and because the absolute optimal values differ across runs.
All code is available at https://github.com/justinpayan/RAU2.

**Overall Performance** Table 2 shows the results for the binarized version of AAMAS 2015 bids. Similar tables for the 5 other settings are included in Appendix G. Each row shows the metrics for the allocation produced by the method that optimizes for the objective shown in the left-most column. All nonrobust methods have robust welfare 0, indicating that if robustness to adversarial noise is desired, it is very important to consider this objective explicitly. Relatively little noise is actually present in this dataset, as the $\mathrm{CVaR}_{0.01}$ is quite high for both the naïve USW-optimal and GESW-optimal in all cases. Since the USW-optimal solution has a very high GESW, we implement a simulated example to explore when the USW-optimal solution fails to have a high GESW. We find that in a number of settings, the GESW of the USW-optimal solution is much lower than that of the GESW-optimal solution. Appendix I explains the details of the simulation setting and the results.

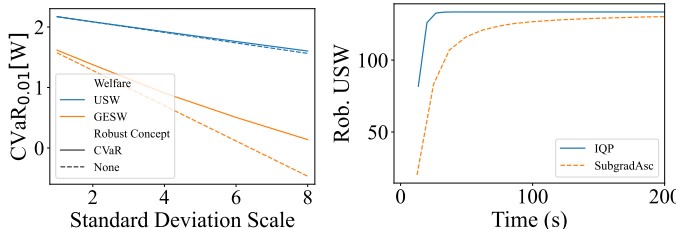

Figure 1: Left: CVaR as noise increases for AAMAS 2015. Right: Convergence behavior of the Iterated Quadratic Program (Iterated QP) vs. Adversarial Projected Subgradient Ascent approach on AAMAS 2015.

Table 2: Performance of different allocations across each metric on the AAMAS 2015 dataset.

| Allocation | Evaluation Objective | | | | | |
| | USW | GESW | CVaR USW | CVaR GESW | Rob. USW | Rob. GESW |
| --- | --- | --- | --- | --- | --- | --- |
| USW | $\mathbf{1.00 \pm 0}$ | $\mathbf{1.00 \pm 0}$ | $\mathbf{1.00 \pm 0}$ | $\mathbf{1.00 \pm 0}$ | $0 \pm 0$ | $0 \pm 0$ |
| GESW | $0.97 \pm 0.01$ | $\mathbf{1.00 \pm 0}$ | $0.97 \pm 0.01$ | $0.97 \pm 0.02$ | $0 \pm 0$ | $0 \pm 0$ |
| CVaR USW | $\mathbf{1.00 \pm 0}$ | $0.99 \pm 0$ | $\mathbf{1.00 \pm 0}$ | $0.99 \pm 0$ | $0 \pm 0$ | $0 \pm 0$ |
| CVaR GESW | $0.98 \pm 0$ | $0.99 \pm 0$ | $0.97 \pm 0.01$ | $\mathbf{1.00 \pm 0}$ | $0 \pm 0$ | $0 \pm 0$ |
| Rob. USW | $0.92 \pm 0.01$ | $0.90 \pm 0.02$ | $0.92 \pm 0.01$ | $0.90 \pm 0.02$ | $\mathbf{1.00 \pm 0}$ | $\mathbf{1.00 \pm 0}$ |
| Rob. GESW | $0.89 \pm 0.04$ | $0.85 \pm 0.06$ | $0.89 \pm 0.04$ | $0.86 \pm 0.06$ | $0.88 \pm 0.02$ | $\mathbf{1.00 \pm 0}$ |

**Robustness under Increasing Uncertainty** Figure 1 shows the $CVaR_{0.01}$ on the Gaussian version of all three datasets as we artificially increase the amount of noise. We multiply the standard deviations of the Gaussian distributions by a scalar and optimize for the CVaR or the naïvely-computed USW and GESW. We then plot $CVaR_{0.01}$ as the noise increases. Although the CVaR approach is less important at low noise levels, the CVaR of welfare decreases for both welfare measures as noise increases. GESW has a sharper decline than USW. We see that as the noise increases, the $CVaR_{0.01}$ of the baseline USW and GESW maximizing allocations drops off relative to the same value for the CVaR-optimized allocation. We also verify that when we model valuations using a negatively-skewed Gaussian distribution with the same means and variances, we see increasing importance of optimizing for CVaR relative to uncertainty-unaware USW and robust USW. The difference is sharper as the skewness parameter gets more negative. Details of this experiment and its results are included in Appendix J.

**Runtime** For the robust optimization setting with ellipsoidal uncertainty sets (derived from confidence intervals over the Gaussian process matrix factorization), we compare the Iterated QP approach (Corollary 3.3) to adversarial projected subgradient ascent on the original max-min problem (as in [16]). We find that Iterated QP converges much faster than the adversarial projected subgradient ascent algorithm on both AAMAS 2015 (Figure 1) and 2016 (Figure 3). Adversarial projected subgradient ascent fails to converge in $1,000$ iterations for the robust GESW objective on all datasets and the USW objective on AAMAS 2021.

## 6 Conclusion

In conclusion, we explore stochastic and robust optimization regimes for utilitarian and group egalitarian welfare objectives. Robust optimization algorithms depend on the form of the uncertainty set. We show that when the uncertainty set has linear constraints only, the resulting problem is an LP and can be solved efficiently. Under ellipsoidal constraints, we demonstrate that iterative quadratic programming approach converges much faster than adversarial projected subgradient ascent. In the stochastic regime, we lay out the sample complexity of CVaR for the utilitarian welfare objective. We demonstrate the feasibility of estimating probability distributions and uncertainty sets on three years of bid data from AAMAS, and show that the robust and CVaR approaches demonstrated in this paper combat the uncertainty present in these three datasets.

## Acknowledgments and Disclosure of Funding

This work is generously supported by Army Research Lab DEVCOM Data and Analysis Center - Contract W911QX23D0009 and NSF grant IIS-2327057. Cousins was supported by the University of Massachusetts Center for Data Science Fellowship. This work was carried out using high performance computing equipment obtained under a grant from the Collaborative R&D Fund managed by the Massachusetts Technology Collaborative.

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

## A   Additional Related Work

Gorissen *et al.* [26] provide an excellent overview of optimization under uncertainty, including techniques used in this work, while Ben-Tal *et al.* [7], Bertsimas *et al.* [8] offer additional background on robust optimization. A standard approach in this regime is analyzing the dual of the uncertainty, as we generally do in this work. Stochastic optimization has a wide literature; the books by Birge and Louveaux [9], Levy *et al.* [37], Prékopa [47], Ruszczyński and Shapiro [52] present wide-ranging introductions to the topic. Conditional value at risk (CVaR) can often be approximately optimized by sampling and optimizing over an objective composing the different samples [35, 40, 50].

## B   Broader Impacts

We believe this work has the potential for a significant positive societal impact. Fair resource allocation algorithms are essential for various systems, including assigning reviewers in peer review processes, allocating resources to homeless and low-income populations, distributing emergency response resources during natural disasters, and resettling refugees. In this work, we develop methods for efficiently optimizing allocations of constrained resources under various fairness objectives while addressing uncertainty in resource preferences. These methods can be directly applied to the aforementioned problems. However, we advise users to conduct extensive testing on similar datasets before deploying these algorithms in real-world scenarios. We also encourage users to perform rigorous causal analysis and controlled experiments to validate their predictive models before applying our methods.

## C   Limitations

The CVaR approach requires solving linear programs with a large number of samples to be effective, which makes them computationally expensive. One potential solution is to leverage importance sampling methods to reduce the variance of the estimator [19, 58]. Future research could benefit from empirically and theoretically analyzing other fairness objectives like Nash welfare [15], Gini index [24], and envy-freeness [39].

## D   Constructing uncertainty sets

In this section we demonstrate a simple and natural approach to construct an uncertainty set using a logistic regression estimator. Logistic regression models with bounded cross-entropy loss result in polyhedral uncertainty sets. Replacing the logistic regression model with a model with bounded squared-error loss, or simply taking the confidence interval of a multivariate Gaussian, results in truncated ellipsoidal uncertainty sets. We construct uncertainty sets per group in all cases.

Assume we have a discrete set of $c$ values $L \subseteq \mathbb{R}$, with $L = \{\ell_1, \ldots \ell_c\}$. For each agent $i$ and item type $j$ we denote the true distribution over values $p^*(\ell|(i,j))$ and the distribution predicted by the logistic regression model is $\hat{p}(\ell|(i,j))$.

We estimate the cross-entropy loss of the model on a test set $T$, where $|T| = t$. This test set can be segmented by the group identities of the agents, such that we have $T_{G_1}, T_{G_2}, \ldots T_{G_g}$ for each of the $g$ groups (with sizes $t_{G_1}, \ldots t_{G_g}$). We assume that the test set comes from the same distribution as the agent-item pairs of the assignment problem; this can be achieved either during dataset construction or by limiting the assignments (through the $C$ constraints) to better reflect the test distribution. We can also apply likelihood reweighting in our uncertainty set construction, as in [16], though we do not do so here.

For an agent $a$ and item type $i$, the cross-entropy loss of the distribution $\hat{p}$ with respect to the distribution $p$ is defined as

$\mathbb{H}(p(\ell|(a,i)), \hat{p}(\ell|(a,i))) \doteq -\sum_{\ell \in L} p(\ell|(a,i)) \ln \hat{p}(\ell|(a,i))$. For each $T_G$, we compute the mean of the cross-entropy loss $\hat{\mu}_G = \frac{1}{t_G} \sum_{(a,i) \in T_G} \mathbb{H}(p(\ell|(a,i)), \hat{p}(\ell|(a,i)))$, as well as the standard error of the mean $\hat{\sigma}_G = \left( \frac{1}{t_G} \sum_{(i,j) \in T_G} (\mathbb{H}(p(\ell|(a,i)), \hat{p}(\ell|(a,i))) - \hat{\mu}_G)^2 \right)^{\frac{1}{2}}$. We model the distribution over cross-entropy losses for group $G$ as $\mathcal{N}(\hat{\mu}_G, \hat{\sigma}_G)$. We want an uncertainty set $\mathcal{V}$ such that

the true values lie outside $\mathcal{V}$ with probability at most $\alpha$. Thus, using a union bound, we require each uncertainty set $\mathcal{V}_G$ for individual groups to contain the true valuations with probability at least $1 - \frac{\alpha}{g}$. We can thus give the bound that the cross entropy loss is at most $\Phi^{-1}(1 - \frac{\alpha}{g}, \hat{\mu}_G, \hat{\sigma}_G)$, where $\Phi^{-1}(p, \mu, \sigma)$ denotes the $p$ percentile of a Gaussian distribution with mean $\mu$ and standard deviation $\sigma$.

In our assignment problem, for each group $G$ with agents $N_G$ we obtain the uncertainty set

$$\frac{1}{t_G m} \sum_{a \in N_G, i \in I} \mathbb{H}(p(\ell|(a,i)), \hat{p}(\ell|(a,i))) \leq \Phi^{-1}(1 - \frac{\alpha}{g}, \hat{\mu}_G, \hat{\sigma}_G) \ .$$

The bound can be made tighter if we restrict some pairs using $C$, in which case the cross-entropy term on the left side is only averaged over the pairs which are not restricted.

## E  Logistic and Gaussian Process Matrix Factorization

In this section we define matrix factorization models using either logistic regression or Gaussian processes. The logistic regression model can be used for predicting the missing elements of the binarized bid matrices, while the Gaussian process model is used for predicting missing real-valued elements. Both models define probability distributions over outcomes, which we use to compute and evaluate the CVaR of utilitarian and egalitarian welfare. For the logistic model, we build a polyhedral uncertainty set by estimating the cross-entropy loss on a held-out test set (as shown in Appendix D), and for the Gaussian process model we simply consider the confidence intervals of the resulting Gaussian distribution.

For the binarized bids, we first set aside some of the observed bids as a test set. We estimate the missing bids and the bids for the held-out test pairs using logistic matrix factorization. Setting a hidden dimension size $d$, we construct two matrices $\mathbf{X} \in \mathbb{R}^{n \times d}$ and $\mathbf{Y} \in \mathbb{R}^{m \times d}$. We set $d = 20$. Let $\mathbf{V}^*$ denote the true binarized bid matrix, where we observe entries for the training set pairs $(a, i) \in T$. We predict the probability of an affirmative bid as $f((\mathbf{X}\mathbf{Y}^\intercal)_{a,i})$ where $f$ is the logistic sigmoid function. We select $\mathbf{X}$ and $\mathbf{Y}$ to minimize the loss function

$$\sum_{(a,i) \in T} -\mathbf{V}^*_{a,i} \ln\left(f((\mathbf{X}\mathbf{Y}^\intercal)_{a,i})\right) - (1 - \mathbf{V}^*_{a,i}) \ln\left(1 - f((\mathbf{X}\mathbf{Y}^\intercal)_{a,i})\right) \ .$$

For CVaR, we take samples from the distribution defined by $f(\mathbf{X}\mathbf{Y}^\intercal)$, assuming all pairs are independently-distributed. We also construct an uncertainty set as described in Appendix D using the cross-entropy loss on the test pairs.

Under the Gaussian process matrix factorization model [36], we simply predict a mean and variance of a Gaussian distribution for each reviewer-paper pair. We can then sample values independently for each pair, or give a confidence interval for the joint Gaussian with $mn - 1$ degrees of freedom.

## F  Grouping Papers and Reviewers

We group papers and reviewers as follows: given the real-valued bids in the set $\{0.01, .5, 1\}$ we set unknown bids to be $0$. We then construct a graph with all reviewers and papers as nodes, and the bid score between reviewers and papers is the edge weight. All inter-reviewer and inter-paper edges are set to $0$ edge weight. We apply spectral embedding with $5$ dimensions to transform the nodes into vectors, and cluster the resulting vectors into $4$ clusters to obtain $4$ groups containing both papers and reviewers. To ensure a balance of reviewers and papers across clusters, we employ Lloyd's algorithm for KMeans clustering with the modification that during each assignment step we enforce a lower bound on the number of papers and number of reviewers assigned to each cluster.

## G  Additional Experiments

For the binarized AAMAS 2016 and 2021 datasets, Tables 3 and 4 show the performance of the baseline USW and GESW maximizing allocations, the $\text{CVaR}_{0.01}$ USW and GESW maximizing allocations, and the robust USW and GESW maximizing allocations at the $\alpha = 0.3$ level. Because

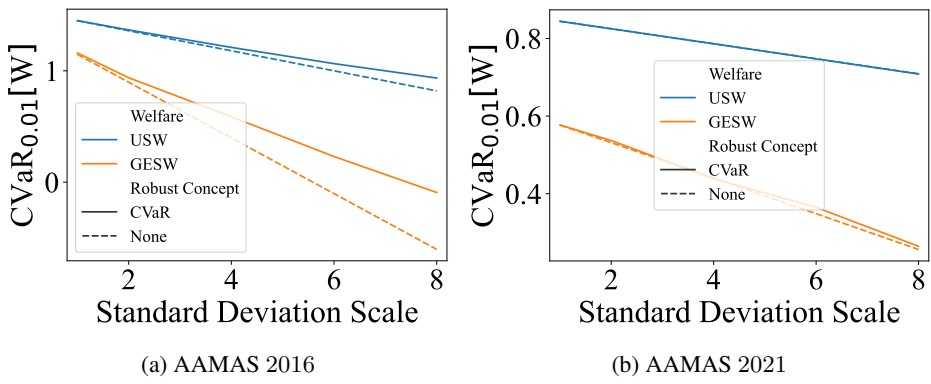

(a) AAMAS 2016           (b) AAMAS 2021

Figure 2: $\text{CVaR}_{0.01}$ as noise increases for AAMAS 2016 and 2021.

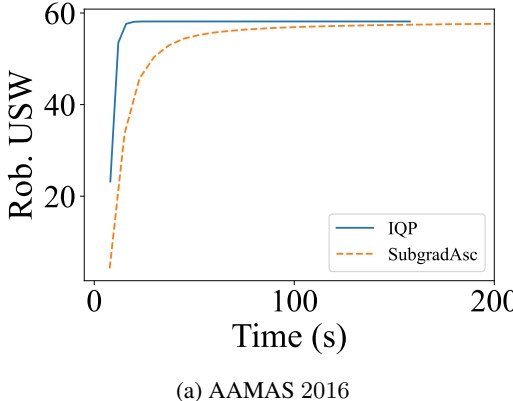

(a) AAMAS 2016

Figure 3: Convergence of the Iterated QP vs. adversarial projected subgradient ascent on AAMAS 2016 dataset for the adversarial USW objective. The Iterated QP (in blue) converges much faster.

so many of the bids in AAMAS 2021 are recorded as `no`, since `no` is the default bid, we randomly select 90% of the `no` bids to be converted to `no response`.

Tables 5 to 7 show the same results for the Gaussian matrix factorization version of the 3 datasets, with the $\text{CVaR}_{0.01}$ estimated by sampling from the estimated Gaussian distribution, and the adversarial welfare computed over the truncated ellipsoidal uncertainty set corresponding to the $1 - \alpha$ confidence interval of the Gaussian.

Table 3: Performance of different allocations across each metric on the AAMAS 2016 dataset.

| Allocation | Evaluation Objective | | | | | |
| | USW | GESW | CVaR USW | CVaR GESW | Rob. USW | Rob. GESW |
|---|---|---|---|---|---|---|
| USW | **1.00 ± 0** | **1.00 ± 0** | **1.00 ± 0** | **1.00 ± 0** | 0 ± 0 | 0 ± 0 |
| GESW | 0.99 ± 0 | **1.00 ± 0** | 0.99 ± 0 | 0.99 ± 0.01 | 0 ± 0 | 0 ± 0 |
| CVaR USW | 0.99 ± 0 | 0.98 ± 0.01 | 0.99 ± 0 | 0.98 ± 0.01 | 0 ± 0 | 0 ± 0 |
| CVaR GESW | 0.99 ± 0.01 | 0.99 ± 0.01 | 0.98 ± 0.01 | **1.00 ± 0** | 0 ± 0 | 0 ± 0 |
| Rob. USW | 0.91 ± 0.02 | 0.87 ± 0.03 | 0.91 ± 0.02 | 0.90 ± 0.03 | **1.00 ± 0** | **1.00 ± 0** |
| Rob. GESW | 0.76 ± 0.05 | 0.66 ± 0.04 | 0.76 ± 0.05 | 0.65 ± 0.05 | 0.74 ± 0.10 | **1.00 ± 0** |

Table 4: Performance of different allocations across each metric on the AAMAS 2021 dataset.

| Allocation | Evaluation Objective | | | | | |
| --- | --- | --- | --- | --- | --- | --- |
| | USW | GESW | CVaR USW | CVaR GESW | Rob. USW | Rob. GESW |
| USW | $\mathbf{1.00 \pm 0}$ | $\mathbf{1.00 \pm 0}$ | $\mathbf{1.00 \pm 0}$ | $\mathbf{1.00 \pm 0}$ | $0 \pm 0$ | $0.40 \pm 0.49$ |
| GESW | $\mathbf{1.00 \pm 0}$ | $\mathbf{1.00 \pm 0}$ | $\mathbf{1.00 \pm 0}$ | $\mathbf{1.00 \pm 0}$ | $0 \pm 0$ | $0.40 \pm 0.49$ |
| CVaR USW | $\mathbf{1.00 \pm 0}$ | $\mathbf{1.00 \pm 0}$ | $\mathbf{1.00 \pm 0}$ | $\mathbf{1.00 \pm 0}$ | $0 \pm 0$ | $0.40 \pm 0.49$ |
| CVaR GESW | $\mathbf{1.00 \pm 0}$ | $\mathbf{1.00 \pm 0}$ | $0.99 \pm 0$ | $\mathbf{1.00 \pm 0}$ | $0 \pm 0$ | $0.40 \pm 0.49$ |
| Rob. USW | $0.85 \pm 0.04$ | $0.69 \pm 0.14$ | $0.84 \pm 0.05$ | $0.64 \pm 0.19$ | $\mathbf{1.00 \pm 0}$ | $\mathbf{1.00 \pm 0}$ |
| Rob. GESW | $0.48 \pm 0.09$ | $0.32 \pm 0.12$ | $0.43 \pm 0.09$ | $0.20 \pm 0.12$ | $0.07 \pm 0.08$ | $\mathbf{1.00 \pm 0}$ |

Table 5: Performance of different allocations across each metric on the Gaussian AAMAS 2015 dataset.

| Allocation | Evaluation Objective | | | | | |
| --- | --- | --- | --- | --- | --- | --- |
| | USW | GESW | CVaR USW | CVaR GESW | Rob. USW | Rob. GESW |
| USW | $\mathbf{1.00 \pm 0}$ | $0.95 \pm 0.03$ | $\mathbf{1.00 \pm 0}$ | $0.94 \pm 0.04$ | $0.61 \pm 0.19$ | $0.34 \pm 0.34$ |
| GESW | $0.87 \pm 0.08$ | $\mathbf{1.00 \pm 0}$ | $0.86 \pm 0.09$ | $0.98 \pm 0.02$ | $0.42 \pm 0.30$ | $0.32 \pm 0.35$ |
| CVaR USW | $\mathbf{1.00 \pm 0}$ | $0.94 \pm 0.03$ | $\mathbf{1.00 \pm 0}$ | $0.96 \pm 0.04$ | $0.63 \pm 0.19$ | $0.35 \pm 0.34$ |
| CVaR GESW | $0.90 \pm 0.06$ | $0.99 \pm 0.01$ | $0.90 \pm 0.07$ | $\mathbf{1.00 \pm 0}$ | $0.51 \pm 0.26$ | $0.36 \pm 0.33$ |
| Rob. USW | $0.86 \pm 0.07$ | $0.76 \pm 0.12$ | $0.88 \pm 0.06$ | $0.80 \pm 0.10$ | $\mathbf{1.00 \pm 0}$ | $0.99 \pm 0.01$ |
| Rob. GESW | $0.75 \pm 0.13$ | $0.77 \pm 0.12$ | $0.76 \pm 0.13$ | $0.82 \pm 0.09$ | $0.87 \pm 0.09$ | $\mathbf{1.00 \pm 0}$ |

Table 6: Performance of different allocations across each metric on the Gaussian AAMAS 2016 dataset.

| Allocation | Evaluation Objective | | | | | |
| --- | --- | --- | --- | --- | --- | --- |
| | USW | GESW | CVaR USW | CVaR GESW | Rob. USW | Rob. GESW |
| USW | $\mathbf{1.00 \pm 0}$ | $0.99 \pm 0.01$ | $\mathbf{1.00 \pm 0}$ | $0.99 \pm 0.02$ | $0.47 \pm 0.27$ | $0.25 \pm 0.38$ |
| GESW | $0.91 \pm 0.06$ | $\mathbf{1.00 \pm 0}$ | $0.91 \pm 0.07$ | $0.98 \pm 0.01$ | $0.37 \pm 0.32$ | $0.24 \pm 0.38$ |
| CVaR USW | $\mathbf{1.00 \pm 0}$ | $0.98 \pm 0.02$ | $\mathbf{1.00 \pm 0}$ | $0.99 \pm 0.01$ | $0.52 \pm 0.25$ | $0.27 \pm 0.37$ |
| CVaR GESW | $0.92 \pm 0.05$ | $0.98 \pm 0.02$ | $0.92 \pm 0.06$ | $\mathbf{1.00 \pm 0}$ | $0.41 \pm 0.31$ | $0.28 \pm 0.37$ |
| Rob. USW | $0.84 \pm 0.08$ | $0.77 \pm 0.12$ | $0.86 \pm 0.07$ | $0.84 \pm 0.09$ | $\mathbf{1.00 \pm 0}$ | $\mathbf{1.00 \pm 0}$ |
| Rob. GESW | $0.73 \pm 0.14$ | $0.76 \pm 0.13$ | $0.74 \pm 0.13$ | $0.84 \pm 0.09$ | $0.85 \pm 0.09$ | $\mathbf{1.00 \pm 0}$ |

Table 7: Performance of different allocations across each metric on the Gaussian AAMAS 2021 dataset.

| Allocation | Evaluation Objective | | | | | |
| --- | --- | --- | --- | --- | --- | --- |
| | USW | GESW | CVaR USW | CVaR GESW | Rob. USW | Rob. GESW |
| USW | $\mathbf{1.00 \pm 0}$ | $\mathbf{1.00 \pm 0.01}$ | $\mathbf{1.00 \pm 0}$ | $\mathbf{1.00 \pm 0.01}$ | $0.53 \pm 0.26$ | $0.21 \pm 0.40$ |
| GESW | $0.80 \pm 0.12$ | $\mathbf{1.00 \pm 0}$ | $0.79 \pm 0.12$ | $0.99 \pm 0.01$ | $0.24 \pm 0.39$ | $0.20 \pm 0.40$ |
| CVaR USW | $\mathbf{1.00 \pm 0}$ | $\mathbf{1.00 \pm 0.01}$ | $\mathbf{1.00 \pm 0}$ | $\mathbf{1.00 \pm 0.01}$ | $0.53 \pm 0.26$ | $0.21 \pm 0.40$ |
| CVaR GESW | $0.85 \pm 0.08$ | $\mathbf{1.00 \pm 0}$ | $0.84 \pm 0.08$ | $\mathbf{1.00 \pm 0}$ | $0.36 \pm 0.34$ | $0.20 \pm 0.40$ |
| Rob. USW | $0.81 \pm 0.11$ | $0.69 \pm 0.16$ | $0.81 \pm 0.11$ | $0.71 \pm 0.16$ | $\mathbf{1.00 \pm 0}$ | $\mathbf{1.00 \pm 0.01}$ |
| Rob. GESW | $0.70 \pm 0.17$ | $0.68 \pm 0.17$ | $0.71 \pm 0.17$ | $0.70 \pm 0.16$ | $0.88 \pm 0.10$ | $\mathbf{1.00 \pm 0}$ |

# H Proofs

## H.1 Proof of Proposition 3.1

**Proposition 3.1** (Robust Utilitarian Welfare Dual)**.** *The problem in* (1) *is equivalent to solving*

$$\max_{\substack{\boldsymbol{\xi} \in \Lambda \\ \boldsymbol{\lambda} \in \mathbb{R}_{0+}^l, \boldsymbol{\beta} \in \mathbb{R}_{0+}^k}} \mathbf{p}^\mathsf{T}\mathbf{T}\mathbf{q} + \boldsymbol{\beta}^\mathsf{T}\mathbf{e} - \frac{1}{4}\mathbf{p}^\mathsf{T}\mathbf{T}\mathbf{p} + \sum_{i=1}^\ell \left(\lambda_i \bar{\mathbf{v}}_i^\mathsf{T}\mathbf{S}_i^{-1}\bar{\mathbf{v}}_i - \lambda_i r_i^2\right) - \mathbf{q}^\mathsf{T}\mathbf{T}\mathbf{q} \ , \tag{3}$$

*where* $\mathbf{p} = \boldsymbol{\xi} - Q^\mathsf{T}\boldsymbol{\beta}$, $\mathbf{q} = \sum_{i=1}^\ell \lambda_i \mathbf{S}_i^{-1}\bar{\mathbf{v}}_i$, *and* $\mathbf{T} = \left(\sum_{i=1}^\ell \lambda_i \mathbf{S}_i^{-1}\right)^{-1}$. *Let* $\boldsymbol{\xi}^*$ *be the optimal* $\boldsymbol{\xi}$ *in* (3). *Then the optimal allocation* $\mathbf{a}^*$ *can be derived from* $\boldsymbol{\xi}^*$ *by finding* $\mathbf{a} \in \mathcal{A}$ *such that* $\mathbf{a} \preceq \boldsymbol{\xi}^*$.

*Proof.* Consider the inner-minimization problem

$$\min_{\mathbf{v} \in \mathbb{R}^{nm}} \sum_{G \in \mathcal{G}} \mathbf{a}_G^\mathsf{T}\mathbf{v}_G$$
$$\forall i \in [1, l] : (\mathbf{v} - \bar{\mathbf{v}}_i)^\mathsf{T}\mathbf{S}_i^{-1}(\mathbf{v} - \bar{\mathbf{v}}_i) \leq r_i^2$$
$$Q\mathbf{v} \succeq \mathbf{e}$$
$$\mathbf{v} \succeq 0 \ .$$

Note that the above optimization problem is convex as the objective is an affine combination of $\mathbf{v}$, which is convex, and the linear and quadratic constraints are also convex. Thus, from the theory of convex optimization (Section 5.1 in Boyd *et al.* [12]), we know that maximizing the dual of a convex optimization problem is equivalent to minimizing its primal counterpart. We will therefore use the Lagrangian method for computing the dual of the above problem.

We will use $\boldsymbol{\beta} \in \mathbb{R}_{0+}^k$ to represent the dual variable corresponding to the linear constraints $Q\mathbf{v} \succeq \mathbf{e}$, $\boldsymbol{\lambda} \in \mathbb{R}_{0+}^\ell$ to represent the dual variable associated with the ellipsoidal constraints, and $\boldsymbol{\zeta} \in \mathbb{R}_{0+}^{nm}$ to represent the dual variable corresponding to the non-negativity constraint on $\mathbf{v}$. The Lagrangian for the above problem is given by

$$L(\mathbf{v}, \boldsymbol{\lambda}, \boldsymbol{\beta}, \boldsymbol{\zeta}) = \mathbf{a}^\mathsf{T}\mathbf{v} + \sum_{i=1}^\ell \lambda_i\left((\mathbf{v} - \bar{\mathbf{v}}_i)^\mathsf{T}\mathbf{S}_i^{-1}(\mathbf{v} - \bar{\mathbf{v}}_i) - r_i^2\right) - \boldsymbol{\beta}^\mathsf{T}(Q\mathbf{v} - \mathbf{e}) - \boldsymbol{\zeta}^\mathsf{T}\mathbf{v} \ . \tag{13}$$

From the first-order optimality conditions, we get

$$\frac{\partial L(\mathbf{v}, \boldsymbol{\lambda}, \boldsymbol{\beta}, \boldsymbol{\zeta})}{\partial \mathbf{v}} = 0$$

$$\mathbf{a} + \sum_{i=1}^\ell 2\lambda_i \mathbf{S}_i^{-1}(\mathbf{v} - \bar{\mathbf{v}}_i) - Q^\mathsf{T}\boldsymbol{\beta} - \boldsymbol{\zeta} = 0$$

$$\implies \mathbf{v} = \left(\sum_{i=1}^\ell 2\lambda_i \mathbf{S}_i^{-1}\right)^{-1}\left(\sum_{i=1}^\ell 2\lambda_i \mathbf{S}_i^{-1}\bar{\mathbf{v}}_i - (\mathbf{a} - Q^\mathsf{T}\boldsymbol{\beta} - \boldsymbol{\zeta})\right) \ .$$

Substituting this value of $\mathbf{v}$ in (13), we get

$$\max_{\substack{\boldsymbol{\lambda} \in \mathbb{R}_{0+}^\ell \\ \boldsymbol{\beta} \in \mathbb{R}_{0+}^{nm} \\ \boldsymbol{\zeta} \in \mathbb{R}_{0+}^{nm}}} -\frac{1}{4}\left((\mathbf{a} - Q^\mathsf{T}\boldsymbol{\beta} - \boldsymbol{\zeta})^\mathsf{T}(\sum_{i=1}^\ell \lambda_i \mathbf{S}_i^{-1})^{-1}(\mathbf{a} - Q^\mathsf{T}\boldsymbol{\beta} - \boldsymbol{\zeta})\right) + \sum_{i=1}^\ell \lambda_i \bar{\mathbf{v}}_i^\mathsf{T}\mathbf{S}_i^{-1}\bar{\mathbf{v}}_i$$
$$-\left(\sum_{i=1}^\ell \lambda_i \mathbf{S}_i^{-1}\bar{\mathbf{v}}_i\right)^\mathsf{T}\left(\sum_{i=1}^\ell \lambda_i \mathbf{S}_i^{-1}\right)^{-1}\left(\sum_{i=1}^\ell \lambda_i \mathbf{S}_i^{-1}\bar{\mathbf{v}}_i\right)$$
$$+ (\mathbf{a} - Q^\mathsf{T}\boldsymbol{\beta} - \boldsymbol{\zeta})^\mathsf{T}\left(\sum_{i=1}^\ell \lambda_i \mathbf{S}_i^{-1}\right)^{-1}\left(\sum_{i=1}^\ell \lambda_i \mathbf{S}_i^{-1}\bar{\mathbf{v}}_i\right) - \sum_{i=1}^\ell \lambda_i r_i^2 + \boldsymbol{\beta}^\mathsf{T}\mathbf{e} \ .$$

Finally, we substitute the above dual problem back into the original problem in (1) to get

$$\max_{\substack{\mathbf{a}\in\mathcal{A} \\ \boldsymbol{\lambda}\in\mathbb{R}^{\ell}_{0+} \\ \boldsymbol{\beta}\in\mathbb{R}^{nm}_{0+} \\ \boldsymbol{\zeta}\in\mathbb{R}^{nm}_{0+}}} -\frac{1}{4}\big((\mathbf{a}-\boldsymbol{Q}^{\intercal}\boldsymbol{\beta}-\boldsymbol{\zeta})^{\intercal}(\sum_{i=1}^{\ell}\lambda_i\mathbf{S}_i^{-1})^{-1}(\mathbf{a}-\boldsymbol{Q}^{\intercal}\boldsymbol{\beta}-\boldsymbol{\zeta})\big) + \sum_{i=1}^{\ell}\lambda_i\bar{\mathbf{v}}_i^{\intercal}\mathbf{S}_i^{-1}\bar{\mathbf{v}}_i$$
$$-(\sum_{i=1}^{\ell}\lambda_i\mathbf{S}_i^{-1}\bar{\mathbf{v}}_i)^{\intercal}(\sum_{i=1}^{\ell}\lambda_i\mathbf{S}_i^{-1})^{-1}(\sum_{i=1}^{\ell}\lambda_i\mathbf{S}_i^{-1}\bar{\mathbf{v}}_i)$$
$$+(\mathbf{a}-\boldsymbol{Q}^{\intercal}\boldsymbol{\beta}-\boldsymbol{\zeta})^{\intercal}(\sum_{i=1}^{\ell}\lambda_i\mathbf{S}_i^{-1})^{-1}(\sum_{i=1}^{\ell}\lambda_i\mathbf{S}_i^{-1}\bar{\mathbf{v}}_i) - \sum_{i=1}^{\ell}\lambda_i r_i^2 + \boldsymbol{\beta}^{\intercal}\mathbf{e} \ .$$

Note that the dual problem is concave in $\boldsymbol{\lambda}$, $\boldsymbol{\beta}$, and $\boldsymbol{\zeta}$ (Section 5.1 in [12]). However, it is unclear if the dual is concave in allocation $\mathbf{a}$. In order to guarantee concavity, we use the change of variables $\boldsymbol{\xi} = \mathbf{a} - \boldsymbol{\zeta}$. From affine-composition rule in convex optimization (Section 3.2.2 in [12]), we know that if $f(\boldsymbol{x})$ given $\boldsymbol{x}\in\mathbb{R}^n$ is convex, then $f(\boldsymbol{A}\boldsymbol{x}+\boldsymbol{b})$ given $\boldsymbol{A}\in\mathbb{R}^{n\times n}, \boldsymbol{b}\in\mathbb{R}^n$ is also convex in $\boldsymbol{x}$. Thus, the variable change $\boldsymbol{\xi} = \mathbf{a} - \boldsymbol{\zeta}$ results in a objective that is concave in $\boldsymbol{\xi}, \boldsymbol{\lambda}, \boldsymbol{\beta}$. The allocation variable $\mathbf{a}$ and the dual variable $\boldsymbol{\zeta}$ only appear in a linear constraint, which is also concave. Thus, the optimization problem is concave in $\mathbf{a}, \boldsymbol{\beta}, \boldsymbol{\lambda}$, and $\boldsymbol{\xi}$, and $\boldsymbol{\zeta}$.

$$\max_{\substack{\boldsymbol{\xi}\in\Lambda,\boldsymbol{\lambda}\in\mathbb{R}^{l}_{0+} \\ \boldsymbol{\beta}\in\mathbb{R}^{k}_{0+} \\ \mathbf{a}\in\mathcal{A},\boldsymbol{\zeta}\in\mathbb{R}^{mn}_{0+}}} \mathbf{p}^{\intercal}\mathbf{T}\mathbf{q} + \boldsymbol{\beta}^{\intercal}\mathbf{e} - \frac{1}{4}\mathbf{p}^{\intercal}\mathbf{T}\mathbf{p} + \sum_{i=1}^{\ell}\Big(\lambda_i\bar{\mathbf{v}}_i^{\intercal}\mathbf{S}_i^{-1}\bar{\mathbf{v}}_i - \lambda_i r_i^2\Big) - \mathbf{q}^{\intercal}\mathbf{T}\mathbf{q} \tag{14}$$
$$\text{s.t. } \boldsymbol{\xi} = \mathbf{a} - \boldsymbol{\zeta} \ ,$$

where $\mathbf{p} = \boldsymbol{\xi} - \boldsymbol{Q}^{\intercal}\boldsymbol{\beta}$, $\mathbf{T} = \big(\sum_{i=1}^{\ell}\lambda_i\mathbf{S}_i^{-1}\big)^{-1}$, $\mathbf{q} = \sum_{i=1}^{\ell}\lambda_i\mathbf{S}_i^{-1}\bar{\mathbf{v}}_i$, and $\Lambda = \mathcal{A} - \mathbb{R}^{nm}_{0+} = \{\boldsymbol{\xi}\in\mathbb{R}^{nm} \mid \forall a\in N : \sum_{i\in I}\xi_{am+i} \le \bar{\kappa}_a, \forall i\in I : \sum_{a\in N}\xi_{am+i} \le \bar{\psi}_i, \boldsymbol{\xi}\preceq c\}$.

We can solve the optimization problem in (14) using standard convex optimization techniques [12].

Alternatively, we can further simplify the problem by eliminating the allocation variable $\mathbf{a}$ and the dual variable $\boldsymbol{\zeta}$ and subsequently deriving them from the solution of the resultant problem.

Note that in (14), $\mathbf{a} - \boldsymbol{\zeta} = \boldsymbol{\xi}$. Let $(\mathbf{a}^*, \boldsymbol{\zeta}^*)$ represent an optimal $(\mathbf{a}, \boldsymbol{\zeta})$ pair for the problem in (14). Now there can be multiple pairs of $(\mathbf{a}, \boldsymbol{\zeta})$ that are optimal. To eliminate $\boldsymbol{\zeta}$ and $\mathbf{a}$, we need to first ensure that there exists a $\boldsymbol{\xi}'\in\Lambda$ such that $\boldsymbol{\xi}' = \mathbf{a}^* - \boldsymbol{\zeta}^*$ for at least one optimal pair $(\mathbf{a}^*, \boldsymbol{\zeta}^*)$. It is easy to see that if there exists such a $\boldsymbol{\xi}'\in\Lambda$, then, $\boldsymbol{\xi}'$ maximizes the objective in (14). Furthermore, since $\Lambda$ is defined solely by upper-bound constraints on $\boldsymbol{\xi}$ imposed by $\mathcal{A}$, we can easily verify that it will contain at least one instance of $\boldsymbol{\xi}'$ that satisfies $\boldsymbol{\xi}' = \mathbf{a}^* - \boldsymbol{\zeta}^*$.

Thus, we can break down the problem in (14) into two sub-problems. In the first problem, we obtain the optimal value of $\boldsymbol{\lambda}$, $\boldsymbol{\xi}$ and $\boldsymbol{\beta}$ by solving

$$\boldsymbol{\zeta}^*, \boldsymbol{\beta}^*, \boldsymbol{\xi}^* = \arg\max_{\substack{\boldsymbol{\zeta}\in\mathbb{R}^{nm}_{0+} \\ \boldsymbol{\beta}\in\mathbb{R}^{k}_{0+} \\ \boldsymbol{\xi}\in\Lambda}} \mathbf{p}^{\intercal}\mathbf{T}\mathbf{q}^{\intercal} + \boldsymbol{\beta}^{\intercal}\mathbf{e} - \frac{1}{4}\mathbf{p}^{\intercal}\mathbf{T}\mathbf{p} + \sum_{i=1}^{\ell}\Big(\lambda_i\bar{\mathbf{v}}_i^{\intercal}\mathbf{S}_i^{-1}\bar{\mathbf{v}}_i - \lambda_i r_i^2\Big) - \mathbf{q}^{\intercal}\mathbf{T}\mathbf{q} \ ,$$

where $\mathbf{p} = \boldsymbol{\xi} - \boldsymbol{Q}^{\intercal}\boldsymbol{\beta}$, $\mathbf{q} = \sum_{i=1}^{\ell}\lambda_i\mathbf{S}_i^{-1}\bar{\mathbf{v}}_i$, and $\mathbf{T} = \big(\sum_{i=1}^{\ell}\lambda_i\mathbf{S}_i^{-1}\big)^{-1}$. Then, we can compute the set of optimal pairs $(\mathbf{a}^*, \boldsymbol{\zeta}^*)$ by solving the system of equations: $\{(\mathbf{a}, \boldsymbol{\zeta}) \mid \mathbf{a}\in\mathcal{A}, \boldsymbol{\zeta}\in\mathbb{R}^{nm}_{0+}, \mathbf{a}\preceq \boldsymbol{\xi}\}$. $\square$

## H.2 Proof of Corollary 3.2

**Corollary 3.2** (Utilitarian Welfare with Polyhedral Uncertainty). *In the case where the uncertainty set $\mathcal{V}$ is defined purely by linear constraints, i.e., $\mathcal{V} = \{\mathbf{v}\in\mathbb{R}^{nm} \mid \boldsymbol{Q}\mathbf{v}\succeq\mathbf{e}, \mathbf{v}\succeq 0\}$, the optimal allocation $\mathbf{a}^*$ for the problem in (1) can be computed by solving the linear program*

$$\max_{\mathbf{a}\in\mathcal{A}, \boldsymbol{\beta}\in\mathbb{R}^{k}_{0+}} \boldsymbol{\beta}^{\intercal}\mathbf{e} \quad \text{s.t. } \boldsymbol{Q}^{\intercal}\boldsymbol{\beta}\preceq\mathbf{a} \ .$$

*Proof.* Now consider the inner-minimization problem:

$$\min_{\mathbf{v}\in\mathbb{R}^{nm}} \mathbf{a}^\mathsf{T}\mathbf{v}$$
$$\boldsymbol{Q}\mathbf{v} \succeq \mathbf{e}$$
$$\mathbf{v} \succeq 0 \ .$$

We compute the dual of the above problem using the Lagrangian method. We will use $\boldsymbol{\beta} \in \mathbb{R}^k_{0+}$ to represent the dual variable corresponding to the linear constraints $\boldsymbol{Q}\mathbf{v} \succeq \mathbf{e}$ and $\boldsymbol{\zeta} \in \mathbb{R}^{nm}_{0+}$ to represent the dual variable corresponding to the non-negativity constraint on $\mathbf{v}$.

$$L(\mathbf{v}, \boldsymbol{\beta}, \boldsymbol{\zeta}) = \mathbf{a}^\mathsf{T}\mathbf{v} - \boldsymbol{\beta}^\mathsf{T}(\boldsymbol{Q}\mathbf{v} - \mathbf{e}) - \boldsymbol{\zeta}^\mathsf{T}\mathbf{v}$$
$$= (\mathbf{a} - \boldsymbol{Q}^\mathsf{T}\boldsymbol{\beta} - \boldsymbol{\zeta})^\mathsf{T}\mathbf{v} + \boldsymbol{\beta}^\mathsf{T}\mathbf{e}$$
$$L(\boldsymbol{\beta}, \boldsymbol{\zeta}) = \begin{cases} \boldsymbol{\beta}^\mathsf{T}\mathbf{e} & (\mathbf{a} - \boldsymbol{Q}^\mathsf{T}\boldsymbol{\beta} - \boldsymbol{\zeta}) \succeq 0 \\ -\infty & \text{otherwise} \end{cases} \ .$$

Therefore, the dual is given by

$$\max_{\substack{\boldsymbol{\beta}\in\mathbb{R}^k_{0+} \\ \boldsymbol{\zeta}\in\mathbb{R}^{nm}_{0+}}} \boldsymbol{\beta}^\mathsf{T}\mathbf{e}$$
$$\boldsymbol{Q}^\mathsf{T}\boldsymbol{\beta} - \boldsymbol{\zeta} \preceq \mathbf{a} \ .$$

Since $\boldsymbol{\zeta}$ is non-negative, we can eliminate it to get

$$\max_{\boldsymbol{\beta}\in\mathbb{R}^k_{0+}} \boldsymbol{\beta}^\mathsf{T}\mathbf{e}$$
$$\boldsymbol{Q}^\mathsf{T}\boldsymbol{\beta} \preceq \mathbf{a} \ .$$

By combining the dual with the outer-maximization problem in (1), we obtain the final result.

$\square$

## H.3  Proof of Corollary 3.3

**Corollary 3.3** (Utilitarian Welfare with Ellipsoidal Uncertainty). *Suppose that the set $\mathcal{V}$ in (1) is defined by a single truncated ellipsoidal constraint, i.e., $\mathcal{V} = \{\mathbf{v} \in \mathbb{R}^{nm} \mid (\mathbf{v} - \bar{\mathbf{v}})^\mathsf{T}\mathbf{S}^{-1}(\mathbf{v} - \bar{\mathbf{v}}) \le r^2, \mathbf{v} \succeq 0\}$. The problem in (1) is equivalent to solving*

$$\max_{\lambda\in\mathbb{R}_{0+}, \boldsymbol{\xi}\in\Lambda} \boldsymbol{\xi}^\mathsf{T}\bar{\mathbf{v}} - \frac{\boldsymbol{\xi}^\mathsf{T}\mathbf{S}\boldsymbol{\xi}}{4\lambda} - \lambda r^2 \ . \tag{4}$$

*The exact optimal solution $(\lambda^*, \boldsymbol{\xi}^*)$ to Equation (4) can be computed by alternately performing two steps until convergence: first, fixing $\boldsymbol{\xi}$ and optimizing $\lambda$, i.e., $\lambda = \sqrt{\boldsymbol{\xi}^\mathsf{T}\mathbf{S}\boldsymbol{\xi}}/2r$, and second, fixing $\lambda$ and solving a concave quadratic program to optimize $\boldsymbol{\xi}$. The optimal allocation $\mathbf{a}^*$ can be computed from $\boldsymbol{\xi}^*$ as in Proposition 3.1.*

*Proof.* Consider the inner-minimization problem

$$\min_{\mathbf{v}\in\mathbb{R}^{nm}} \sum_{G\in\mathcal{G}} \mathbf{a}_G^\mathsf{T}\mathbf{v}_G$$
$$(\mathbf{v} - \bar{\mathbf{v}})^\mathsf{T}\mathbf{S}^{-1}(\mathbf{v} - \bar{\mathbf{v}}) \le r^2$$
$$\mathbf{v} \succeq 0 \ .$$

Similar to the approach in Proposition 3.1, we will use the Lagrangian method for computing the dual of the above problem. We will use $\lambda \in \mathbb{R}_{0+}$ to represent the dual variable associated with the ellipsoidal constraint, and $\boldsymbol{\zeta} \in \mathbb{R}^{nm}_{0+}$ to represent the dual variable corresponding to the non-negativity constraint on $\mathbf{v}$. The Lagrangian for the above problem is given by

$$L(\mathbf{v}, \lambda, \boldsymbol{\zeta}) = \mathbf{a}^\mathsf{T}\mathbf{v} + \lambda\left((\mathbf{v} - \bar{\mathbf{v}})^\mathsf{T}\mathbf{S}^{-1}(\mathbf{v} - \bar{\mathbf{v}}) - r^2\right) - \boldsymbol{\zeta}^\mathsf{T}\mathbf{v} \ . \tag{15}$$

From the first-order optimality conditions, we get

$$\frac{\partial L(\mathbf{v}, \lambda, \boldsymbol{\zeta})}{\partial \mathbf{v}} = 0$$

$$\mathbf{a} + 2\lambda \mathbf{S}^{-1}(\mathbf{v} - \bar{\mathbf{v}}) - \boldsymbol{\zeta} = 0$$

$$\implies \mathbf{v} = \left(\frac{\mathbf{S}}{2\lambda}\right)\left(2\lambda \mathbf{S}^{-1}\bar{\mathbf{v}} - (\mathbf{a} - \boldsymbol{\zeta})\right) .$$

Substituting the value of $\mathbf{v}$ in the above equation in (15), we get

$$\max_{\substack{\lambda \in \mathbb{R}_{0+} \\ \boldsymbol{\zeta} \in \mathbb{R}_{0+}^{nm}}} -\frac{1}{4}\left((\mathbf{a} - \boldsymbol{\zeta})^{\mathsf{T}}\frac{\mathbf{S}}{\lambda}(\mathbf{a} - \boldsymbol{\zeta})\right) + (\mathbf{a} - \boldsymbol{\zeta})^{\mathsf{T}}\bar{\mathbf{v}} - \lambda r^2 .$$

From the theory of convex optimization [12], we know that the dual of a convex optimization problem is always concave, and therefore, the above optimization problem is concave in $\lambda$ and $\boldsymbol{\zeta}$. Combining the dual with the outer-maximization problem in (1), we get

$$\max_{\mathbf{a} \in \mathcal{A}} \max_{\substack{\lambda \in \mathbb{R}_{0+}^{\ell} \\ \boldsymbol{\zeta} \in \mathbb{R}_{0+}^{nm}}} -\frac{1}{4}\left((\mathbf{a} - \boldsymbol{\zeta})^{\mathsf{T}}\frac{\mathbf{S}}{\lambda}(\mathbf{a} - \boldsymbol{\zeta})\right) + (\mathbf{a} - \boldsymbol{\zeta})^{\mathsf{T}}\bar{\mathbf{v}} - \lambda r^2 .$$

To further obtain concavity in allocation $\mathbf{a}$, we follow the same procedure as in Proposition 3.1 and use the change of variables $\boldsymbol{\xi} = \mathbf{a} - \boldsymbol{\zeta}$ to get

$$\max_{\substack{\mathbf{a} \in \mathcal{A} \\ \lambda \in \mathbb{R}_{0+} \\ \boldsymbol{\xi} \in \Lambda}} \boldsymbol{\xi}^{\mathsf{T}}\bar{\mathbf{v}} - \frac{\boldsymbol{\xi}^{\mathsf{T}}\mathbf{S}\boldsymbol{\xi}}{4\lambda} - \lambda r^2 \tag{16}$$

$$\text{s.t. } \boldsymbol{\xi} = \mathbf{a} - \boldsymbol{\zeta} ,$$

where $\Lambda$ is defined as in (2).

As a result of the change of variables, the allocation variable $\mathbf{a}$ and the dual variable $\boldsymbol{\zeta}$ now appear only in a linear constraint, which is convex. Furthermore, due to the affine composition property of convex functions (Section 3.2.2 in Boyd *et al.* [12]), the objective remains concave in $\lambda$ and $\boldsymbol{\xi}$. Thus, the above optimization problem is concave in $\mathbf{a}$, $\lambda$, $\boldsymbol{\zeta}$, and $\boldsymbol{\xi}$.

Similar to the approach used in the proof of Proposition 3.1, we further simplify the problem by eliminating the allocation variables $\mathbf{a}$ and the dual variable $\boldsymbol{\zeta}$ and subsequently deriving them from the solution of the resultant problem.

From (16), we know that that $\mathbf{a} - \boldsymbol{\zeta} = \boldsymbol{\xi}$. Let $(\mathbf{a}^*, \boldsymbol{\zeta}^*)$ represent an optimal $(\mathbf{a}, \boldsymbol{\zeta})$ pair for the problem in (14). Note that there can be multiple pairs of $(\mathbf{a}, \boldsymbol{\zeta})$ that are optimal. To eliminate $\boldsymbol{\zeta}$ and $\mathbf{a}$, we need to find a set of feasible $\boldsymbol{\xi}$, which we denote by $\Lambda$, such that there exists a $\boldsymbol{\xi}' \in \Lambda$ such that $\boldsymbol{\xi}' = \mathbf{a}^* - \boldsymbol{\zeta}^*$ for at least one optimal pair $(\mathbf{a}^*, \boldsymbol{\zeta}^*)$. It is easy to see that if there exists such a $\boldsymbol{\xi}' \in \Lambda$, then, $\boldsymbol{\xi}'$ maximizes the objective in (16). Furthermore, it easy to verify that $\Lambda$ satisfies this criteria for optimality, i.e., it contains at least one $\boldsymbol{\xi}'$ that satisfies $\boldsymbol{\xi}' = \mathbf{a}^* - \boldsymbol{\zeta}^*$, for some optimal pair $(\mathbf{a}^*, \boldsymbol{\zeta}^*)$.

Thus, we can efficiently solve the problem in (16) in two steps. First, we obtain the optimal value of $\lambda$ and $\boldsymbol{\xi}$ by solving the problem

$$\lambda^*, \boldsymbol{\xi}^* = \arg\max_{\substack{\lambda \in \mathbb{R}_{0+} \\ \boldsymbol{\xi} \in \Lambda}} \boldsymbol{\xi}^{\mathsf{T}}\bar{\mathbf{v}} - \frac{\boldsymbol{\xi}^{\mathsf{T}}\mathbf{S}\boldsymbol{\xi}}{4\lambda} - \lambda r^2 .$$

As in Proposition 3.1, we can compute the set of optimal pairs $(\mathbf{a}^*, \boldsymbol{\zeta}^*)$ by solving the system of equations: $\{(\mathbf{a}, \boldsymbol{\zeta}) \mid \mathbf{a} \in \mathcal{A}, \boldsymbol{\zeta} \in \mathbb{R}_{0+}^{nm}, \mathbf{a} - \boldsymbol{\zeta} = \boldsymbol{\xi}\}$.

$\square$

## H.4 Proof of Proposition 3.5

**Proposition 3.5** (Robust Group Egalitarian Dual). *The problem in (5) is equivalent to solving*

$$
\max_{\substack{\boldsymbol{\xi}\in\Lambda \\ \boldsymbol{\lambda}\in\mathbb{R}_{0+}^{g\times l} \\ \boldsymbol{\beta}\in\mathbb{R}_{0+}^{g\times k}}} \min_{G\in\mathcal{G}} \boldsymbol{\beta}_G^\mathsf{T}\mathbf{e}_G + \mathbf{p}_G^\mathsf{T}\mathbf{T}_G\mathbf{q}_G - \frac{1}{4}\mathbf{p}_G^\mathsf{T}\mathbf{T}_G\mathbf{p}_G +
\sum_{i=1}^{\ell}\left(\lambda_{G,i}\bar{\mathbf{v}}_{G,i}^\mathsf{T}\mathbf{T}_G\bar{\mathbf{v}}_{G,i} - \lambda_{G,i}r_{G,i}^2\right) - \mathbf{q}_G^\mathsf{T}\mathbf{T}_G\mathbf{q}_G \ , \tag{7}
$$

*where for each group* $G\in\mathcal{G}$, $\mathbf{p}_G = \boldsymbol{\xi}_G - \boldsymbol{Q}_G^\mathsf{T}\boldsymbol{\beta}_G$, $\mathbf{q}_G = \sum_{i=1}^{\ell}\lambda_{G,i}\mathbf{S}_{G,i}^{-1}\bar{\mathbf{v}}_{G,i}$, $\mathbf{T}_G = \left(\sum_{i=1}^{\ell},\lambda_{G,i}\mathbf{S}_{G,i}^{-1}\right)^{-1}$, *and* $\Lambda$ *is defined as in Equation* (2). *The optimal allocation* $\mathbf{a}^*$ *can be computed from* $\boldsymbol{\xi}^*$ *as in Proposition 3.1.*

*Proof.* Consider the following optimization problem.

$$
\max_{\mathbf{a}\in\mathcal{A}} \min_{G\in\mathcal{G}} \min_{\mathbf{v}_G\in\mathcal{V}_G} \mathbf{a}_G^\mathsf{T}\mathbf{v}_G
$$
$$
\forall i \in [1,l] : \forall G\in\mathcal{G} : (\mathbf{v}_G - \bar{\mathbf{v}}_{i,G})^\mathsf{T}\mathbf{S}_{i,G}^{-1}(\mathbf{v}_G - \bar{\mathbf{v}}_{i,G}) \leq r_{i,G}^2 \tag{17}
$$
$$
\forall G\in\mathcal{G} : Q_G\mathbf{v}_G \succeq \mathbf{e}_G
$$
$$
\mathbf{v}_G \succeq 0 \ .
$$

It is important to note that the inner-most minimization is a convex optimization problem and the outer-maximization is a concave maximization problem. This is due to the fact that affine functions are either concave or convex and minimum of concave objectives is concave.

Notice that the inner-most minimization problem for each group is independent of other groups. Thus, we can simply replace each of these minimization problems with their Lagrangian dual counterparts. Furthermore, we note that these duals are computed following the approach outlined in the proof of Proposition 3.1 and are exact equivalents of their respective primal counterparts [12]. The resultant optimization problem is given by

$$
\max_{\mathbf{a}\in\mathcal{A}} \min_{G\in\mathcal{G}} \max_{\substack{\boldsymbol{\lambda}_G\in\mathbb{R}_{0+}^\ell \\ \boldsymbol{\beta}_G\in\mathbb{R}_{0+}^k \\ \boldsymbol{\zeta}_G\in\mathbb{R}_{0+}^{|G|m}}} -\frac{1}{4}\left( \left(\mathbf{a}_G - \boldsymbol{Q}_G^\mathsf{T}\boldsymbol{\beta}_G - \boldsymbol{\zeta}_G\right)^\mathsf{T}\left(\sum_{i=1}^{\ell}\lambda_i\mathbf{S}_{G,i}^{-1}\right)^{-1}\left(\mathbf{a}_G - \boldsymbol{Q}_G^\mathsf{T}\boldsymbol{\beta}_G - \boldsymbol{\zeta}_G\right)\right) +
$$
$$
\sum_{i=1}^{\ell}\lambda_{G,i}\bar{\mathbf{v}}_{G,i}^\mathsf{T}\mathbf{S}_{G,i}^{-1}\bar{\mathbf{v}}_{G,i} - \left(\sum_{i=1}^{\ell}\lambda_{G,i}\mathbf{S}_{G,i}^{-1}\bar{\mathbf{v}}_{G,i}\right)^\mathsf{T}\left(\sum_{i=1}^{\ell}\lambda_{G,i}\mathbf{S}_{G,i}^{-1}\right)^{-1}\left(\sum_{i=1}^{\ell}\lambda_{G,i}\mathbf{S}_{G,i}^{-1}\bar{\mathbf{v}}_{G,i}\right)
$$
$$
+ (\mathbf{a}_G - \boldsymbol{Q}_G^\mathsf{T}\boldsymbol{\beta}_G - \boldsymbol{\zeta}_G)^\mathsf{T}\left(\sum_{i=1}^{\ell}\lambda_{G,i}\mathbf{S}_{G,i}^{-1}\right)^{-1}\left(\sum_{i=1}^{\ell}\lambda_{G,i}\mathbf{S}_{G,i}^{-1}\bar{\mathbf{v}}_{G,i}\right)
$$
$$
- \sum_{i=1}^{\ell}\lambda_{G,i}r_{G,i}^2 + \boldsymbol{\beta}_G^\mathsf{T}\mathbf{e}_G \ .
$$

Notice that the inner-most maximization problem for each group $G$ is concave in $\boldsymbol{\lambda}_G$, $\boldsymbol{\beta}_G$, and $\boldsymbol{\zeta}_G$. We now apply the same procedure as in the proof of Proposition 3.1 to obtain concavity in allocation $\mathbf{a}$. Using the change of variables $\forall G\in\mathcal{G} : \boldsymbol{\xi}_G = \mathbf{a}_G - \boldsymbol{\zeta}_G$, we get

$$
\max_{\mathbf{a}\in\mathcal{A}} \min_{G\in\mathcal{G}} \max_{\substack{\boldsymbol{\lambda}_G\in\mathbb{R}^l \\ \boldsymbol{\beta}_G\in\mathbb{R}_{0+}^k \\ \boldsymbol{\zeta}_G\in\mathbb{R}_{0+}^{|G|m} \\ \boldsymbol{\xi}_G\in\mathbb{R}^{|G|m}}} \boldsymbol{\beta}_G^\mathsf{T}\mathbf{e}_G + \mathbf{p}_G^\mathsf{T}\mathbf{T}_G\mathbf{q}_G - \frac{1}{4}\mathbf{p}_G^\mathsf{T}\mathbf{T}_G\mathbf{p}_G + \sum_{i=1}^{\ell}\left(\lambda_{G,i}\bar{\mathbf{v}}_{G,i}^\mathsf{T}\mathbf{T}_G\bar{\mathbf{v}}_{G,i} - \lambda_{G,i}r_{G,i}^2\right) - \mathbf{q}_G^\mathsf{T}\mathbf{T}_G\mathbf{q}_G
$$
$$
\text{s.t. } \boldsymbol{\xi}_G = \mathbf{a}_G - \boldsymbol{\zeta}_G \ ,
$$

where for any $G\in\mathcal{G}, \mathbf{p}_G = \boldsymbol{\xi}_G - \boldsymbol{Q}_G^\mathsf{T}\boldsymbol{\beta}_G$, $\mathbf{T}_G = \left(\sum_{i=1}^{\ell}\lambda_{G,i}\mathbf{S}_{G,i}^{-1}\right)^{-1}$, and $\mathbf{q}_G = \sum_{i=1}^{\ell}\lambda_{G,i}\mathbf{S}_{G,i}^{-1}\bar{\mathbf{v}}_{G,i}$.

Since the inner maximization for each group is independent of the other groups, we can re-order the inner minimization over groups and the inner-maximization problem. Thus, without loss of

generality, we can write the above optimization problem as

$$\max_{\substack{\mathbf{a}\in\mathcal{A} \\ \boldsymbol{\zeta}\in\mathbb{R}^{nm} \\ \boldsymbol{\lambda}\in\mathbb{R}_{0+}^{g\times l} \\ \boldsymbol{\beta}\in\mathbb{R}_{0+}^{g\times k} \\ \boldsymbol{\xi}\in\mathbb{R}^{nm}}} \min_{G\in\mathcal{G}} \boldsymbol{\beta}_G^{\mathsf{T}}\mathbf{e}_G + \mathbf{p}_G^{\mathsf{T}}\mathbf{T}_G\mathbf{q}_G - \frac{1}{4}\mathbf{p}_G^{\mathsf{T}}\mathbf{T}_G\mathbf{p}_G + \sum_{i=1}^{\ell}\left(\lambda_{G,i}\bar{\mathbf{v}}_{G,i}^{\mathsf{T}}\mathbf{T}_G\bar{\mathbf{v}}_{G,i} - \lambda_{G,i}r_{G,i}^2\right) - \mathbf{q}_G^{\mathsf{T}}\mathbf{T}_G\mathbf{q}_G$$

$$\text{s.t. } \boldsymbol{\xi}_G = \mathbf{a}_G - \boldsymbol{\zeta}_G \ .$$

Using the same technique as in the proof of Proposition 3.1, we can simplify the problem by eliminating the variables $\mathbf{a}$ and $\boldsymbol{\zeta}$ in the above problem and then derive them from the optimal $\boldsymbol{\xi}$.

Eliminating $\boldsymbol{\zeta}$ and $\mathbf{a}$ in the above equation, we get

$$\boldsymbol{\lambda}^*,\boldsymbol{\beta}^*,\boldsymbol{\xi}^*$$

$$= \arg\max_{\substack{,\boldsymbol{\lambda}\in\mathbb{R}_{0+}^{g\times l}, \\ \boldsymbol{\beta}\in\mathbb{R}_{0+}^{g\times k} \\ \boldsymbol{\xi}\in\Lambda}} \min_{G\in\mathcal{G}} \boldsymbol{\beta}_G^{\mathsf{T}}\mathbf{e}_G + \mathbf{p}_G^{\mathsf{T}}\mathbf{T}_G\mathbf{q}_G - \frac{1}{4}\mathbf{p}_G^{\mathsf{T}}\mathbf{T}_G\mathbf{p}_G + \sum_{i=1}^{\ell}\left(\lambda_{G,i}\bar{\mathbf{v}}_{G,i}^{\mathsf{T}}\mathbf{T}_G\bar{\mathbf{v}}_{G,i} - \lambda_{G,i}r_{G,i}^2\right) - \mathbf{q}_G^{\mathsf{T}}\mathbf{T}_G\mathbf{q}_G \ ,$$

where for each group $G \in \mathcal{G}$, $\mathbf{p}_G = (\boldsymbol{\xi}_G - \boldsymbol{Q}_G^{\mathsf{T}}\boldsymbol{\beta}_G)$, $\mathbf{q}_G = \sum_{i=1}^{\ell}\lambda_{G,i}\mathbf{S}_{G,i}^{-1}\bar{\mathbf{v}}_{G,i}$, $\mathbf{T}_G = \left(\sum_{i=1}^{\ell}, \lambda_{G,i}\mathbf{S}_{G,i}^{-1}\right)^{-1}$, and $\Lambda = \mathcal{A} - \mathbb{R}_{0+}^{nm} = \{\boldsymbol{\xi}\in\mathbb{R}^{nm} \mid \forall a\in N : \sum_{i\in I}\xi_{am+i} \leq \bar{\kappa}_a, \forall i \in I : \sum_{a\in N}\xi_{am+i} \leq \bar{\psi}_i, \boldsymbol{\xi}\preceq\mathbf{c}\}$.

As in Proposition 3.1, we can determine the set of optimal pairs $(\mathbf{a}^*,\boldsymbol{\zeta}^*)$ by solving the system of equations: $\{(\mathbf{a},\boldsymbol{\zeta}) \mid \mathbf{a}\in\mathcal{A}, \boldsymbol{\zeta}\in\mathbb{R}_{0+}^{nm}, \mathbf{a} - \boldsymbol{\zeta} = \boldsymbol{\xi}\}$.

□

## H.5   Proof of Corollary 3.6

**Corollary 3.6** (Group Egalitarian Welfare with Polyhedral Uncertainty). *In the case where the uncertainty set $\mathcal{V}$ is defined only by linear constraints, i.e., $\mathcal{V} = \{\mathbf{v} \in \mathbb{R}^{nm} \mid \forall G \in \mathcal{G} : \boldsymbol{Q}_G\mathbf{v}_G \succeq \mathbf{e}_G, \mathbf{v} \succeq 0\}$, the max-min-min problem in* (5) *transforms into a linear program.*

*Proof.* Now consider the inner-minimization problem:

$$\max_{\mathbf{a}\in\mathcal{A}} \min_{G\in\mathcal{G}} \min_{\mathbf{v}_G\in\mathbb{R}_{0+}^{|G|m}} \mathbf{a}_G^{\mathsf{T}}\mathbf{v}_G$$

$$\forall G \in \mathcal{G} : Q_G\mathbf{v}_G \succeq \mathbf{e}_G \ .$$

We can compute the Lagrangian dual of the inner-most minimization problem for each group independently by following steps outlined in the proof of Corollary 3.2. Note that since these minimization problems are simple linear programs, their corresponding duals are exact equivalents of their primal counterparts (Section 5.1 in Boyd *et al.* [12]). By substituting the duals in the above problem and reordering the inner-maximization and minimization problems, we obtain

$$\max_{\substack{\mathbf{a}\in\mathcal{A} \\ \boldsymbol{\beta}\in\mathbb{R}^{g\times k}}} \min_{G\in\mathcal{G}} \boldsymbol{\beta}_G^{\mathsf{T}}\mathbf{e}_G$$

$$\boldsymbol{Q}_G^{\mathsf{T}}\boldsymbol{\beta}_G \preceq \mathbf{a}_G \ .$$

Using simple algebraic manipulations, we can further simplify the above optimization problem as

$$\max_{\substack{\mathbf{a}\in\mathcal{A} \\ \boldsymbol{\beta}\in\mathbb{R}^{g\times k} \\ t\in\mathbb{R}}} t$$

$$\forall G \in \mathcal{G} : t \leq \boldsymbol{\beta}_G^{\mathsf{T}}\mathbf{e}_G$$

$$\forall G \in \mathcal{G} : \boldsymbol{Q}_G^{\mathsf{T}}\boldsymbol{\beta}_G \preceq \mathbf{a}_G \ .$$

□

## H.6 Proof of Corollary 3.7

**Corollary 3.7** (Group Egalitarian Welfare with Ellipsoidal Uncertainty)**.** *Suppose that the set $\mathcal{V}$ in (5) is defined by a single truncated ellipsoidal constraint per group i.e., $\mathcal{V} = \{\mathbf{v} \in \mathbb{R}^{nm} \mid \forall G \in \mathcal{G} : (\mathbf{v}_G - \bar{\mathbf{v}}_G) \mathbf{S}_G^{-1}(\mathbf{v}_G - \bar{\mathbf{v}}_G) \leq r_G^2, \mathbf{v} \succeq 0\}$. Then the problem in (5) is equivalent to solving*

$$\max_{\substack{\boldsymbol{\lambda} \in \mathbb{R}_{0+}^g \\ \boldsymbol{\xi} \in \Lambda}} \min_{G \in \mathcal{G}} \boldsymbol{\xi}_G^\intercal \bar{\mathbf{v}}_G - \frac{\boldsymbol{\xi}_G^\intercal \mathbf{S}_G \boldsymbol{\xi}_G}{4\lambda_G} - \lambda_G r_G^2 \ .$$

*The exact optimal solution $(\boldsymbol{\lambda}^*, \boldsymbol{\xi}^*)$ to Equation (4) can be computed by alternately performing two steps until convergence: first, fixing $\boldsymbol{\xi}$ and optimizing $\boldsymbol{\lambda}$, i.e., $\forall G \in \mathcal{G}$, $\lambda_G = \sqrt{\boldsymbol{\xi}_G^\intercal \mathbf{S}_G \boldsymbol{\xi}_G}/2r_G$, and second, fixing $\boldsymbol{\lambda}$ and solving a concave quadratic program to optimize $\boldsymbol{\xi}$. The optimal allocation $\mathbf{a}^*$ can be computed from $\boldsymbol{\xi}^*$ as in Proposition 3.1.*

*Proof.* Consider the following optimization problem.

$$\max_{\mathbf{a} \in \mathcal{A}} \min_{G \in \mathcal{G}} \min_{\mathbf{v}_G \in \mathbb{R}_{0+}^{|G|m}} \mathbf{a}_G^\intercal \mathbf{v}_G$$

$$\forall G \in \mathcal{G} : (\mathbf{v}_G - \bar{\mathbf{v}}_G)^\intercal \mathbf{S}_G^{-1}(\mathbf{v}_G - \bar{\mathbf{v}}_G) \leq r_G^2$$

$$\mathbf{v}_G \succeq 0 \ .$$

Similar to the general version of the problem in (17), the inner-most minimization is a convex optimization problem and the outer-maximization is a concave maximization problem. This is again due to the fact that affine functions are either concave or convex and minimum of concave objectives is concave.

The inner-most minimization over the uncertainty set of valuation matrices is independent for each group. Therefore, by simply replacing each of these minimization problems with their respective Lagrangian duals, as computed in Corollary 3.3, we obtain

$$\max_{\mathbf{a} \in \mathcal{A}} \min_{G \in \mathcal{G}} \max_{\substack{\lambda_G \in \mathbb{R}_{0+} \\ \boldsymbol{\zeta}_G \in \mathbb{R}_{0+}^{|G|m}}} -\frac{1}{4\lambda_G} \left( (\mathbf{a}_G - \boldsymbol{\zeta}_G)^\intercal \mathbf{S}_G (\mathbf{a}_G - \boldsymbol{\zeta}_G) \right) + (\mathbf{a}_G - \boldsymbol{\zeta}_G)^\intercal \bar{\mathbf{v}}_G - \lambda_G r_G^2 \ .$$

Note that the dual is computed following the approach outlined in the proof of Proposition 3.1.

Using the change of variables $\forall G \in \mathcal{G} : \boldsymbol{\xi}_G = \mathbf{a}_G - \boldsymbol{\zeta}_G$, we get

$$\max_{\mathbf{a} \in \mathcal{A}} \min_{G \in \mathcal{G}} \max_{\substack{\lambda_G \in \mathbb{R}_{0+} \\ \boldsymbol{\zeta}_G \in \mathbb{R}_{0+}^{|G|m} \\ \boldsymbol{\xi}_G \in \mathbb{R}^{|G|m}}} \boldsymbol{\xi}_G^\intercal \bar{\mathbf{v}}_G - \frac{\boldsymbol{\xi}_G^\intercal \mathbf{S}_G \boldsymbol{\xi}_G}{4\lambda_G} - \lambda_G r_G^2 \quad \text{s.t. } \boldsymbol{\xi}_G = \mathbf{a}_G - \boldsymbol{\zeta}_G \ .$$

Since the inner maximization for each group is independent of the other groups, we can re-order the inner minimization over groups and the inner-maximization problem. Thus, without loss of generality, we can write the above optimization problem as

$$\max_{\substack{\mathbf{a} \in \mathcal{A} \\ \boldsymbol{\lambda} \in \mathbb{R}_{0+}^g \\ \boldsymbol{\zeta} \in \mathbb{R}_{0+}^{nm} \\ \boldsymbol{\xi} \in \mathbb{R}^{nm}}} \min_{G \in \mathcal{G}} \boldsymbol{\xi}_G^\intercal \bar{\mathbf{v}}_G - \frac{\boldsymbol{\xi}_G^\intercal \mathbf{S}_G \boldsymbol{\xi}_G}{4\lambda_G} - \lambda_G r_G^2$$

$$\text{s.t. } \boldsymbol{\xi}_G = \mathbf{a}_G - \boldsymbol{\zeta}_G \ .$$

Using the same technique as in the proof of Proposition 3.1, we can simplify the problem by eliminating the variables $\mathbf{a}$ and $\boldsymbol{\zeta}$ in the above problem and then derive them from the optimal $\boldsymbol{\xi}$.

Eliminating $\boldsymbol{\zeta}$ and $\mathbf{a}$ in the above optimization problem, we get

$$\boldsymbol{\lambda}^*, \boldsymbol{\xi}^* = \arg\max_{\substack{\boldsymbol{\lambda} \in \mathbb{R}_{0+}^g \\ \boldsymbol{\xi} \in \Lambda}} \min_{G \in \mathcal{G}} \boldsymbol{\xi}_G^\intercal \bar{\mathbf{v}}_G - \frac{\boldsymbol{\xi}_G^\intercal \mathbf{S}_G \boldsymbol{\xi}_G}{4\lambda_G} - \lambda_G r_G^2 \ ,$$

where $\Lambda = \mathcal{A} - \mathbb{R}_{0+}^{nm} = \{\boldsymbol{\xi} \in \mathbb{R}^{nm} \mid \forall a \in N : \sum_{i \in I} \xi_{am+i} \leq \bar{\kappa}_a, \forall i \in I : \sum_{a \in N} \xi_{am+i} \leq \bar{\psi}_i, \boldsymbol{\xi} \preceq \mathbf{c}\}$.

As in Proposition 3.1, we can now determine the set of optimal pairs $(\mathbf{a}^*, \boldsymbol{\zeta}^*)$ by solving the system of equations: $\{(\mathbf{a}, \boldsymbol{\zeta}) \mid \mathbf{a} \in \mathcal{A}, \boldsymbol{\zeta} \in \mathbb{R}_{0+}^{nm}, \mathbf{a} - \boldsymbol{\zeta} = \boldsymbol{\xi}\}$. $\qquad\square$

## H.7 Proof of Proposition 3.8

**Proposition 3.8** (Decomposition for Monotonic Welfare Functions). *Consider an optimization problem of the form*

$$\max_{\mathbf{a}\in\mathcal{A}} \min_{\mathbf{v}\in\mathcal{V}} \ W_M(\mathbf{u}(\mathbf{a},\mathbf{v})) \ , \tag{8}$$

*where the welfare function $W_M$ is monotonic in the utility of groups. If Assumption 3.4 holds, then (8) simplifies to*

$$\max_{\mathbf{a}\in\mathcal{A}} \ W_M \left( \min_{\mathbf{v}_{G_1}\in\mathcal{V}_{G_1}} u_{G_1}(\mathbf{a}_{G_1},\mathbf{v}_{G_1}), \min_{\mathbf{v}_{G_2}\in\mathcal{V}_{G_2}} u_{G_2}(\mathbf{a}_{G_2},\mathbf{v}_{G_2}),\ldots, \min_{\mathbf{v}_{G_g}\in\mathcal{V}_{G_g}} u_{G_g}(\mathbf{a}_{G_g},\mathbf{v}_{G_g}) \right) \ .$$

*Proof.* The result directly follows from the monotonic property of the welfare function and the independence of the uncertainty sets across groups. □

## H.8 Proof of Proposition 4.1

**Proposition 4.1** (Approximate CVaR of USW). *Given $h$ i.i.d samples of $\tilde{\mathbf{v}}$, i.e., $\mathbf{v}^1,\mathbf{v}^2,\mathbf{v}^3,\ldots,\mathbf{v}^h$ from $\mathcal{D}_{\tilde{\mathbf{v}}}$, the optimal allocation for the problem in (9) can be approximately computed by solving*

$$\max_{\mathbf{a}\in\mathcal{A}} \max_{\substack{\boldsymbol{y}\in\mathbb{R}_{0+}^h \\ b\in\mathbb{R}}} \left( b - \frac{1}{\alpha}\sum_{j=1}^{h} \boldsymbol{y}_j \right) \qquad \forall j\in[1,h]:\ \boldsymbol{y}_j \geq \frac{1}{h}\left( b - \mathbf{w}\cdot\mathbf{u}(\mathbf{a},\mathbf{v}^j) \right) \ . \tag{10}$$

*Proof.* Consider the CVaR of utilitarian welfare optimization problem, given by

$$\max_{\mathbf{a}\in\mathcal{A}} \text{CVaR}_\alpha\left[\mathbf{w}\cdot\mathbf{u}(\mathbf{a},\tilde{\mathbf{v}})\right] \doteq \max_{\mathbf{a}\in\mathcal{A},b\in\mathbb{R}} \left\{ b - \frac{1}{\alpha}\mathop{\mathbb{E}}_{\tilde{\mathbf{v}}\sim\mathcal{D}_{\tilde{\mathbf{v}}}}\left[\left(b-\mathbf{w}\cdot\mathbf{u}(\mathbf{a},\tilde{\mathbf{v}})\right)_+\right] \right\} \ ,$$

Substituting the expectation with the empirical expectation computed from the $h$ samples of random valuation $\tilde{\mathbf{v}}$, and using a slack variable $\boldsymbol{y} \in \mathbb{R}_{0+}^h$ to convert $\left(b-\mathbf{w}\cdot\mathbf{u}(\mathbf{a},\tilde{\mathbf{v}})\right)_+$ term in the expectation to linear constraints, we get

$$\max_{\mathbf{a}\in\mathcal{A}} \max_{\boldsymbol{y}\in\mathbb{R}^h,b\in\mathbb{R}} \left( b - \frac{1}{\alpha}\sum_{j=1}^{h} y_j \right)$$

$$\forall j\in[1,h]:\ y_j \geq 0$$

$$\forall j\in[1,h]:\ y_j \geq \frac{1}{h}\left( b - \sum_{G\in\mathcal{G}} \frac{\mathbf{w}_G}{|G|}\cdot\mathbf{a}_G^\intercal\mathbf{v}_G^j \right)$$

$$\mathbf{a}\in\mathcal{A} \ .$$

□

**Assumption H.1** (L.A. *et al.* [35]). *The random variable $\tilde{x}$ is continuous with probability density function $f$ that satisfies the following condition: There exists $\eta,\gamma>0$ such that $\forall y\in[v_\alpha-\gamma,v_\alpha+\gamma]:\ f(y)>\eta$, where $v_\alpha = F^{-1}(\alpha)$.*

**Theorem H.2** (Theorem 3.1 in [35]). *Let $(\tilde{x}_i)_{i=1}^h$ be a sequence of i.i.d random variables. Suppose that $\tilde{x}_i, i=1,\ldots,n$ are $\sigma-$sub-Gaussian and Assumption H.1 holds. If $c_\alpha$ and $\hat{c}_{h,\alpha}$ represent the true CVaR and the empirical CVaR of random variable $\tilde{x}$ estimated from $h$ samples at confidence level $\alpha$ respectively, then for any $\varepsilon>0$, we have*

$$\Pr\left[|\hat{c}_{h,\alpha}-c_\alpha|>\varepsilon\right] \leq 6\exp\left( \frac{-h\alpha^2\min(\varepsilon^2,16\gamma^2)\min(\eta^2,1)}{8\max(8,\sigma^2)} \right) \ .$$

## H.9 Proof of Proposition 4.2

**Proposition 4.2** (Sample Complexity of Approximate CVaR of USW). *Suppose that $\tilde{\mathbf{v}}$ is a multi-variate sub-Gaussian random variable with mean $\bar{\mathbf{v}} \in \mathbb{R}^{nm}$ and covariance proxy $\mathbf{S} \in \mathbb{R}^{nm \times nm}$, i.e., for all vectors $\mathbf{z} \in \mathbb{R}^{nm}$ : $\mathbb{E}_{\tilde{\mathbf{v}} \sim \mathcal{D}_{\tilde{\mathbf{v}}}} \left[ \exp((\tilde{\mathbf{v}} - \bar{\mathbf{v}})^{\mathsf{T}} \mathbf{z})) \right] \leq \exp(\mathbf{z}^{\mathsf{T}} \mathbf{S} \mathbf{z} / 2)$, and that, for any risk level $\alpha \in (0, \frac{1}{2})$ and allocation $\mathbf{a} \in \mathcal{A}$, there exists probability density threshold $\eta > 0$ and radius $\gamma > 0$, s.t., $f_{\mathbf{a}}(x) > \eta$, $\forall x \in [\nu_\alpha - \gamma, \nu_\alpha + \gamma]$. Set $\forall G \in \mathcal{G} : \mathbf{a}'_G = \frac{\mathbf{w}_G \cdot \mathbf{a}_G}{|G|}$. Then, for any confidence parameter $\delta \in (0, 1)$ and error tolerance $\varepsilon > 0$,*

$$\Pr \left[ \sup_{\mathbf{a} \in \mathcal{A}} |\hat{c}_{h,\alpha}(\mathbf{a}) - c_\alpha(\mathbf{a})| \leq \varepsilon \right] \geq 1 - \delta \quad for \quad h > \left\lceil \frac{8 \max \left( \max_{\mathbf{a} \in \mathcal{A}} \mathbf{a}'^{\mathsf{T}} \mathbf{S} \mathbf{a}', 8 \right) \ln \left( \frac{6|\mathcal{A}|}{\delta} \right)}{\min(\varepsilon^2, 16\gamma^2) \alpha^2 \min(\eta^2, 1)} \right\rceil .$$

*Proof.* From the assumption, we know that the valuation vector $\tilde{\mathbf{v}}$ is a sub-Gaussian random variable that satisfies the following condition

$$\forall \mathbf{z} \in \mathbb{R}^{nm} : \mathbb{E}_{\tilde{\mathbf{v}} \sim \mathcal{D}_{\tilde{\mathbf{v}}}} \left[ \exp((\tilde{\mathbf{v}} - \bar{\mathbf{v}})^{\mathsf{T}} \mathbf{z})) \right] \leq \exp(\mathbf{z}^{\mathsf{T}} \mathbf{S} \mathbf{z} / 2) .$$

Using the above properties of sub-Gaussian, we can establish that the utilitarian welfare for a given allocation $\mathbf{a}$ is also a sub-Gaussian with variance-proxy $= \mathbf{a}'^{\mathsf{T}} \mathbf{S} \mathbf{a}'$ where $\forall G \in \mathcal{G} : \mathbf{a}'_G = \frac{\mathbf{w}_G \cdot \mathbf{a}}{|G|}$.

For any allocation $\mathbf{a}$, let $\hat{c}_{h,\alpha}(\mathbf{a})$ represent the empirical estimate of CVaR of utilitarian welfare and $c_\alpha(\mathbf{a})$ represent the corresponding true value.

Then, using Theorem H.2, we can bound the error of approximating the CVaR of the utilitarian welfare for allocation $\mathbf{a}$ as

$$\Pr \left[ |\hat{c}_{h,\alpha}(\mathbf{a}) - c_\alpha(\mathbf{a})| > \varepsilon \right] \leq 6 \exp \left( \frac{-h\alpha^2 \min(\varepsilon^2, 16\gamma^2) \min(\eta^2, 1)}{8 \max(8, \mathbf{a}'^{\mathsf{T}} \mathbf{S} \mathbf{a}')} \right) .$$

The approximation error for all allocations can be upper-bounded using a union bound as

$$\Pr \left[ \sup_{\mathbf{a} \in \mathcal{A}} |\hat{c}_{h,\alpha}(\mathbf{a}) - c_\alpha(\mathbf{a})| > \varepsilon \right] \leq \sum_{\mathbf{a} \in \mathcal{A}} \Pr \left[ |\hat{c}_{h,\alpha}(\mathbf{a}) - c_\alpha(\mathbf{a})| > \varepsilon \right]$$
$$\leq |\mathcal{A}| 6 \exp \left( \frac{-h\alpha^2 \min(\varepsilon^2, 4\gamma^2) \min(\eta^2, 1)}{8 \max(8, \mathbf{a}'^{\mathsf{T}} \mathbf{S} \mathbf{a}')} \right) . \tag{18}$$

To obtain confidence guarantee $\Pr[\forall \mathbf{a} \in \mathcal{A} : |\hat{c}_{h,\alpha}(\mathbf{a}) - c_\alpha(\mathbf{a})| \leq \varepsilon] \geq 1 - \delta$, we set the R.H.S of (18) to be less than $\delta$, i.e.,

$$|\mathcal{A}| 6 \exp \left( \frac{-h\alpha^2 \min(\varepsilon^2, 4\gamma^2) \min(\eta^2, 1)}{8 \max(8, \mathbf{a}'^{\mathsf{T}} \mathbf{S} \mathbf{a}')} \right) < \delta .$$

Solving for $h$, we get

$$h > \left\lceil \left( \frac{8 \max(\max_{\mathbf{a} \in \mathcal{A}} \mathbf{a}'^{\mathsf{T}} \mathbf{S} \mathbf{a}', 8) \ln \left( \frac{6|\mathcal{A}|}{\delta} \right)}{\min(\varepsilon^2, 4\gamma^2) \alpha^2 \min(\eta^2, 1)} \right) \right\rceil .$$

$\square$

## H.10 Proof of Proposition 4.3

**Proposition 4.3** (CVaR of USW for Gaussian Distributions). *If $\tilde{\mathbf{v}}$ is distributed as a multivariate Gaussian, i.e., $\tilde{\mathbf{v}} \sim \mathcal{N}(\bar{\mathbf{v}}, \mathbf{S})$, then, the optimization problem in (9) simplifies to*

$$\max_{\mathbf{a} \in \mathcal{A}} \mathbf{a}'^{\mathsf{T}} \bar{\mathbf{v}} - \frac{\phi(\Phi^{-1}(\alpha))}{\alpha} \sqrt{\mathbf{a}'^{\mathsf{T}} \mathbf{S} \mathbf{a}'} , \tag{11}$$

*Proof.* The proof simply follows from the fact that for any Gaussian distributed random variable $\tilde{x} \sim \mathcal{N}(\mu, \sigma^2)$ with mean $\mu \in \mathbb{R}$ and standard deviation $\sigma \in \mathbb{R}_{0+}$, $\mathrm{CVaR}[\tilde{x}] = \mu - \frac{\phi(\Phi^{-1}(\alpha))}{\alpha}\sigma$. The mean of utilitarian welfare under the assumption that $\tilde{\mathbf{v}}$ is Gaussian distributed is given by $\mathbf{a'}^\mathsf{T}\bar{\mathbf{v}}$ and variance is given by $\mathbf{a'}^\mathsf{T}\mathbf{S}\mathbf{a'}$, where for any $G \in \mathcal{G}, \mathbf{a}'_G = \frac{\mathbf{w}_G \cdot \mathbf{a}}{|G|}$. Substituting these values in the previously mentioned expression of CVaR of a Gaussian random variable, we obtain the results stated above. $\square$

### H.11 Linear Program for CVaR of Egalitarian Welfare

**Proposition H.3** (Approximate CVaR of GESW). *Given $h$ samples of $\tilde{\mathbf{v}}$, i.e., $\mathbf{v}^1, \mathbf{v}^2, \mathbf{v}^3, \ldots \mathbf{v}^h$ sampled from the valuation distribution $\mathcal{D}_{\tilde{\mathbf{v}}}$, the optimal allocation for the problem in (12) can be approximately computed by solving*

$$\max_{\mathbf{a} \in \mathcal{A}} \max_{\mathbf{y} \in \mathbb{R}^h_{0+}, b \in \mathbb{R}} \left( b - \frac{1}{\alpha} \sum_{j=1}^{h} y_j \right)$$

$$\forall j \in [1, h] : \forall G \in \mathcal{G} : \ y_j \geq \frac{1}{h} \left( b - \frac{1}{|G|} \cdot \mathbf{a}_G^\mathsf{T} \mathbf{v}_G^j \right) \ . \tag{19}$$

*Proof.* Consider the CVaR of egalitarian welfare optimization problem, given by

$$\max_{\mathbf{a} \in \mathcal{A}} \mathrm{CVaR}_\alpha \left[ \min_{G \in \mathcal{G}} \frac{1}{|G|} \cdot \mathbf{a}_G^\mathsf{T} \tilde{\mathbf{v}}_G \right] \quad \dot{=} \max_{b \in \mathbb{R}, \mathbf{a} \in \mathcal{A}} \left\{ b - \frac{1}{\alpha} \mathop{\mathbb{E}}_{\tilde{\mathbf{v}} \sim \mathcal{D}_{\tilde{\mathbf{v}}}} \left[ \left( b - \min_{G \in \mathcal{G}} \frac{1}{|G|} \cdot \mathbf{a}_G^\mathsf{T} \tilde{\mathbf{v}}_G \right)_+ \right] \right\} \ .$$

Substituting the expectation in the above problem with the empirical expectation computed from the $h$ samples of the valuation matrices, we get

$$\approx \max_{b \in \mathbb{R}, \mathbf{a} \in \mathcal{A}} \left\{ b - \frac{1}{\alpha h} \sum_{i=1}^{h} \left( b - \min_{G \in \mathcal{G}} \frac{1}{|G|} \cdot \mathbf{a}_G^\mathsf{T} \mathbf{v}_G^i \right)_+ \right\} \ .$$

Introducing slack variables $\mathbf{y} \in \mathbb{R}^h$, we can write the above problem as

$$\max_{\mathbf{a} \in \mathcal{A}} \max_{\mathbf{y} \in \mathbb{R}^h, b \in \mathbb{R}} \left( b - \frac{1}{\alpha} \sum_{j=1}^{h} y_j \right)$$

$$\forall j \in [1, h] : \ y_j \geq 0$$

$$\forall j \in [1, h] : \ y_j \geq \frac{1}{h} \left( b - \min_{G \in \mathcal{G}} \frac{1}{|G|} \cdot \mathbf{a}_G^\mathsf{T} \mathbf{v}_G^j \right) \ . \tag{20}$$

Without loss of generality, we can represent the above problem as

$$\max_{\mathbf{a} \in \mathcal{A}} \max_{\mathbf{y} \in \mathbb{R}^h, b \in \mathbb{R}} \left( b - \frac{1}{\alpha} \sum_{j=1}^{h} y_j \right)$$

$$\forall j \in [1, h] : \ y_j \geq 0$$

$$\forall j \in [1, h] : \forall G \in \mathcal{G} : \ y_j \geq \frac{1}{h} \left( b - \frac{1}{|G|} \cdot \mathbf{a}_G^\mathsf{T} \mathbf{v}_G^j \right) \ . $$

$\square$

## I  GESW vs. USW under Different Scenarios

We run an experiment to discover scenarios where the USW-optimal solution has very sub-optimal GESW. Using the AAMAS 2015 dataset, we set all of the papers to be group 1, and we create a second, synthetic group of papers by copying and modifying a random subset of the papers. For these papers, we divide the copied valuations by some number, and set to zero all but the top valuations

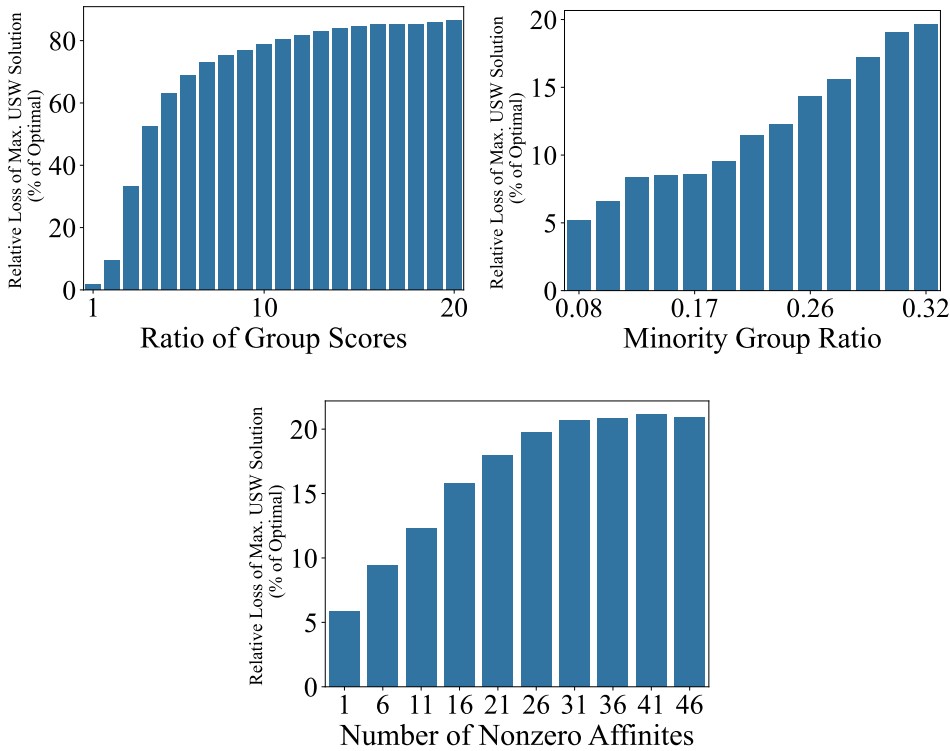

Figure 4: Relative loss (in GESW) of the maximum USW solution, compared to the optimal GESW solution. Results are reported for a synthetic 2-group example, varying 1) the divisor applied to artificially scale the minority group's valuations, 2) the ratio of the minority group to the overall number of papers, and 3) the number of valuations per paper that are artificially set to 0.

per paper. For each setting we compute the percentage of relative loss in GESW incurred by the maximum USW solution. Figure 4 shows the effects of varying the percentage of papers in the minority group, the number of non-zero entries, and the divisor. We use default values of 2 for the divisor, 5 for the number of nonzero entries, and 150 for the size of the minority group (corresponds to a ratio of roughly 20%). Taken together, the results suggest that if one group of papers has overall lower bids than another group, this has a very strong negative effect on the GESW of the USW-optimal solution.

## J  CVaR **Performance under Skewed Gaussians**

We use the AAMAS 2015 dataset and sample each valuation independently from a skewed-Gaussian distribution with varying skew parameter. We use the means and variances estimated by the Gaussian Process matrix factorization model described in Appendix E. The univariate skewed-Gaussian distribution with mean $\mu$, variance $\sigma^2$, and skew $\alpha$ is defined by the probability density function

$$p(\tilde{x}; \alpha) = 2\phi(\tilde{x} - \mu)\Phi(\alpha(\tilde{x} - \mu)/\sigma),$$

where $\phi$ and $\Phi$ are the probability density function and cumulative density function of a standard Gaussian distribution [6].

We compare against the uncertainty-unaware and robust USW solution (using the ellipsoid derived from the Gaussian distribution, which is an optimistic uncertainty set). We optimize and evaluate for $\text{CVaR}_{0.3}$. Table 8 displays the results. As the skew parameter $\alpha$ gets more negative, the $\text{CVaR}_{0.3}$ of welfare of the naïve and robust solutions get much worse compared to the CVaR-optimized solution.

We also consider a constructed scenario with two agents and four items, where each agent must be assigned one item. Each agent's valuation for an item is independent and follows a skewed Gaussian

Table 8: Performance of different allocations for CVaR on the AAMAS 2015 with skewed Gaussians.

| Skew | CVaR USW | USW | Rob. USW |
|------|----------|------|----------|
| $-0.5$ | 1.64 | 1.56 | 1.01 |
| $-1$ | 1.45 | 1.21 | 0.86 |
| $-2$ | 1.33 | 0.96 | 0.75 |
| $-5$ | 1.29 | 0.84 | 0.70 |
| $-10$ | 1.28 | 0.82 | 0.52 |

Table 9: Performance of different allocations for CVaR of USW on toy example with skewed Gaussians.

| Approach | Solution | $\text{CVaR}_{0.04}[\text{USW}]$ |
|----------|----------|----------------------------------|
| Naïve | $\begin{bmatrix} 0 & 0 & 0 & 1 \\ 0 & 0 & 1 & 0 \end{bmatrix}$ | 0.947 |
| Robust | $\begin{bmatrix} 0 & 1 & 0 & 0 \\ 1 & 0 & 0 & 0 \end{bmatrix}$ | 0.972 |
| CVaR | $\begin{bmatrix} 0 & 0 & 1 & 0 \\ 1 & 0 & 0 & 0 \end{bmatrix}$ | **0.985** |

distribution. The mean valuations for the items are represented by the matrix

$$\begin{bmatrix} 0.39 & 0.49 & 0.51 & 0.53 \\ 0.52 & 0.51 & 0.53 & 0.54 \end{bmatrix}.$$

The item values have standard deviations $\begin{bmatrix} 0.01 & 0.04 & 0.05 & 0.09 \end{bmatrix}$ respectively for both agents, and the skew parameter is $\alpha = 5$ for all item-agent pairs. We sample $20,000$ samples from the skewed Gaussian distibutions. We compute the naïve USW- or GESW-optimal allocations using the mean valuations, and we enumerate all possible allocations to identify the CVaR and robust allocations at a confidence level of $0.04$. To estimate the robust statistic at a $0.04$ confidence level, we assume independence and reject any samples that fall outside the $0.04/8$ confidence interval upper/lower bounds for any individual entry. We then compute the minimum welfare on the remaining samples. The computed solutions and the $\text{CVaR}_{0.04}$ of each are displayed in Tables 9 and 10.

In both the USW and GESW cases, the naïve approach selects items 3 and 4, as they have the highest mean values, but it does not account for the uncertainty in the preferences. The robust approach, being more conservative, chooses items 1 and 2 due to their lower uncertainty. The CVaR approach strikes a balance between these two methods, selecting items 1 and 3. Item 1 has a lower mean value with low uncertainty, while item 3 has a higher mean value but with slightly higher uncertainty than item 2.

Table 10: Performance of different allocations for CVaR of GESW on toy example with skewed Gaussians.

| Approach | Solution | $\text{CVaR}_{0.04}[\text{GESW}]$ |
|----------|----------|-----------------------------------|
| Naïve | $\begin{bmatrix} 0 & 0 & 0 & 1 \\ 0 & 0 & 1 & 0 \end{bmatrix}$ | 0.45 |
| Robust | $\begin{bmatrix} 0 & 1 & 0 & 0 \\ 1 & 0 & 0 & 0 \end{bmatrix}$ | 0.46 |
| CVaR | $\begin{bmatrix} 0 & 0 & 1 & 0 \\ 1 & 0 & 0 & 0 \end{bmatrix}$ | **0.47** |

# K  Machine Specification

All experiments were run on Xeon E5-2680 v4 @ 2.40GHz machines with 128GB RAM with each experiment consuming at most 32 GB of memory. We ran 1500 experiments in total and each experiment took 3-4 hours.

