# OpenReview forum: "Fair and Welfare-Efficient Constrained Multi-Matchings under Uncertainty"
_NeurIPS.cc/2024/Conference — NeurIPS 2024 poster_

### Official Review · Reviewer_UQiJ · 2024-06-30

**Soundness:** 4
**Presentation:** 3
**Contribution:** 3
**Rating:** 7
**Confidence:** 4

**Summary:**

The authors propose group-fair multi-matching algorithms in the presence of uncertainty about the valuations of agent-item pairs. The algorithms fall under two families: Conditional Value at Risk (CVaR), and robust optimization. The types of utilities they consider are also two-fold: normal utilitarian, and egalitarian group-fairness. The latter seeks to maximize the utility of the group with the minimum utility.

The authors begin with the robust optimization approach for overall utility maximization, where they derive max-min guarantees for the worst possible valuation within a particular valuation uncertainty set. Here, they propose solving a natural nested concave-convex program. Due to computational feasibility, they would like to utilize the structure of the specific problem in order to achieve a more efficient problem framing. To do this, they demonstrate that the nested program (eq 1) can be re-written as a single concave cubic program. Furthermore, the optimal assignment can be found by solving a particular system of equations with the optimal dual variables (Proposition 3.1). They also show that under some simplified assumptions on the uncertainty set containing the true valuation matrix, the optimal solution can be found by simply solving a LP or QP (Prop. 3.2, 3.3).

The authors then move to robust optimization for egalitarian welfare. Here, the problem is more complex since the objective function loses its smoothness, and is now a nested max min min optimization problem. They make a key assumption on the independence of the uncertainty set w.r.t. the data groups, which helps them swap the minimum over groups and minimum valuation over uncertainty sets. By another dual derivation, one can solve the resulting problem with iterated quadratic programming (Prop. 3.4). Furthermore, the problem has the nice property that under simplified uncertainty sets, it still reduces to solving a single LP (Prop. 3.5). The authors also note that the problem can be decomposed in the case of monotonic welfare functions.

The next topic covered is w.r.t. the CVaR approach, which assumes that the distribution over agent-item valuations is known to the mechanism designer. We begin with the optimal utilitarian assignment. Here, the optimal solution can be computed, but may be computationally infeasible for arbitrary distribution over valuation matrices $D$. Therefore, the authors adopt and approach where they sample many valuation matrices from the distribution, and solve a particular linear program which computes an empirical estimate of the \alpha-CVaR. Prior work shows that this is a consistent estimator as the number of samples goes to infinity, but they provide a concrete sample complexity bound for subgaussian distributions (Prop. 4.3), under assumption 4.2. When the distribution is normal, only the mean and covariance matter, and a particular QP can be solved (Prop. 4.4). Similar results are discussed for the group egalitarian assignment, in section 4.2.

The authors close with a series of experiments on real AAMAS reviewer bid data, where they are able to demonstrate the effectiveness and scalability of their method to a modest sample size. Here, uncertainty is defined by a model fit to predict reviewer bids on unseen examples.

**Strengths:**

The problem considered by the authors, namely, understanding the role of uncertainty in matching algorithms, is important and well-motivated. Furthermore, their setup is extremely practical and testable. I enjoyed reading about the optimization tricks and problem specific structure used to solve each of the instances. I found the fact that the optimal assignment was given by a particular system of linear equations with the optimal dual variables very neat.

Overall, I do not think that any other work in the literature has explored and proposed feasible algorithms for matching in the presence of uncertainty in such a variety of settings (robust optimization under both utility optimal and egalitarian matchings, similarly for CVaR). I also appreciate that the code for running all programs was included. It can be quite difficult to implement programs of this complexity in practice (which a skim of the implemented gurobi programs can verify). I encourage the authors to consider cleaning up and releasing the code as a package, there is certainly a dearth of available code for efficient matching algorithms online currently!

**Weaknesses:**

I believe that there are numerous opportunities to improve the quality of presentation in the main paper. If addressed, I plan to raise my score. Most importantly, Propositions 3.4, 3.3, 3.1 should — in my opinion — be stated informally in the main text. I don’t believe the reader will gain much by staring at the (impressive!) equations. Having the informal version retain the key takeaway — we can reduce to solving a system of equations with the optimal dual variables — would make it much easier to parse, and also emphasize the important (and cool) part of the proposition.

In lines 25-30, you specify where uncertainty may arise from. I would also consider mentioning how you capture this uncertainty: namely, by modeling uncertainty over agent-item pair valuations. These valuations completely capture the “preferences” you discuss later in the paragraph. However, when I think of matchings and preferences, concepts like stability come to mind, which is a much harder constraint to satisfy than utility maximization.

I detail more areas for improving clarity and preciseness of writing in the questions section of my response. Most are minor besides simplifying the main statement of the propositions.

If accepted, I also recommend utilizing the additional space to discuss the experiments in further detail. In particular, is table 2 reporting normalized utilities? Why are most of the results 1.0? Is this to be expected?

**Questions:**

Q1. 113 “always the case that the constraints define a finite set such that |A| ∈ N.” Do you include the possibility that |A| = 0?

Q2. 116 “Let u : A × $\mathcal{V}$ × G → R be an affine function mapping from allocations to utilities for each group.” I don't think $\mathcal{V}$ was defined before this statement, but am assuming that it refers to the set of all valuation matrices. However, in 131, you define $\mathcal{V}$ to be the uncertainty set containing the true valuation matrix with high probability. There is probably a cleaner way to get around this without overloading, since it seems the uncertainty set definition is more important for your discussion throughout. Perhaps simply define the utility function to map from an arbitrary collection of valuations to R, instead of the set of all valuations?

Q3. 133 Not sure whether $W({u}(a))$ formally type checks properly, shouldn’t $u$ be a function of $V$ as well as the assignment $a$ (and also groups, as defined above?)

Q4. 134 $D_v$ is a distribution over valuation matrices? This seems to be defined in the next paragraph 136-145. Is it possible to switch the order of these two paragraphs?

Q5. 184 can probably do without the $\forall G \in \mathcal{G}$ in the sentence immediately following $w_G$. Also, I believe that the presentation would be enhanced by considering numbers to be not-bold (e.g. $w_G$, $u_{G_1}$, etc.), and to keep vectors bold (which the authors have already done). This would certainly help parse the notation, since there is quite a bit.

Q6. 224: valuation matrices of each group are independent of each other. What are the practical implications of this assumption? What kind of problems are we not capturing by making this assumption? This should be stated clearly, and perhaps the assumption should be stated more formally.

Q7. Equation (9), should the LHS term have an overset of $\tilde{v} \sim D_v$ for the CVaR_\alpha term? My understanding is that CVaR is w.r.t. a particular distribution $D$, not a particular draw of the random variable. Are you sweeping this into the CVaR term somehow?

Q8. 326:  “Section 5.1”, should this be table 2?

Q9. How is adversarial welfare defined? The worst welfare within the uncertainty set?

Q10. I do not believe that $W_{USW}$ or $W_{GESW}$ appear in the main paper, and hence, you may be able to remove these acronyms from section 2.1. Furthermore, throughout the paper, you seem to not use the acronyms until presenting the experimental results (which makes sense).

Q11. 339: “Although the CVaR approach is less important at low noise levels, the CVaR of welfare decreases for both welfare measures as noise increases”. As someone not super familiar with CVaR, why is it less important at lower noise levels? Do we expect the noise to mainly impact the right tail or something like that?
Q12. 178-180: rounding algorithms in the case of fractional solutions. Do you think you can expand a bit on how these randomized rounding algorithms apply to your specific experiments? How much of a difference do they make? Is there a Birkhoff-von-Neumann decomposition-type result for randomized-multi-matchings which can be used to sample?

**Limitations:**

The limitations are discussed in the main body of the paper, the most salient of which is that that CVaR requires solving a large number of linear programs to obtain the desired guarantees, and hence can be slow in practice. I don't see this as that big of a problem, however, especially since in the proposed setting of reviewer assignment, only a single matching is constructed and utilized.

---

> ### Author Rebuttal · Authors · 2024-08-06
>
> Thank you for your detailed comments and helpful suggestions.
>
> Please see our replies below.
>
> **Propositions 3.4, 3.3, 3.1 should be stated informally in the main text.**
>
> Thank you for the suggestion. We agree that the equations make it less readable.
> We will replace them with a informal statements describing the propositions, and move the formal statements to the appendix.
>
> **Mention how you capture this uncertainty.**
>
> Thank you for the suggestion. We will incorporate this in the main paper.
>
> **Why are most of the results 1.0? Is this to be expected?**
>
> We briefly mentioned on line 329 objective values are normalized by dividing by the maximum value of that objective per seed.” Table 2 reports means and standard errors over 5 runs. For each run, we execute 6 algorithms (the 6 rows in the table), and evaluate each of those 6 allocations on all 6 objective values. So we get five 6x6 matrices. For each run, we normalize each column by dividing all entries by the max value, so the maximum value is 1.0. We then take the element-wise means and standard errors of those 6x6 matrices over all 5 runs. We normalize in this manner because (a) the absolute optimal welfare differs across runs, and (b) we wanted to highlight the fact that the diagonal is expected to be (weakly) better than the off-diagonal in each run.
>
> **Do you include the possibility that |A| = 0?**
>
> Yes, our proposed techniques can handle this special case. However, we observe that in almost all allocation problems, we would have constraints on the number of items that can/must be assigned to an agent.
>
> **There is probably a cleaner way to get around this without overloading.**
>
> You’re right, this is overloaded. Thank you! We can just replace that with [0,1]^{n \times m}.
>
> **Shouldn’t $u$ be a function of as well as the assignment and also groups, as defined above?**
>
> We would like to clean up this notation a bit in general. It is cleaner to write that $\bf u$ is a vector-valued function giving the welfare for all groups under an allocation and valuation matrix, $\mathbf{u}: \mathcal A \times [0, 1]^{n \times m} \to \mathbb R^g$. On line 133 we will write $W(\mathbf u(\mathbf a, \mathbf v))$.
>
>
> **Are you sweeping this into the CVaR term somehow?**
>
> Yes, that’s right. We were “sweeping this into the CVaR term” as you say, but it would be much clearer if we include the overset. In fact, this is the same distribution over $\tilde{v}$ as in the $\mathbf{E}\_{\tilde{v} \sim D_v}$ operator on the RHS, so it should be written on the LHS too. We will do the same for the $\mathrm{CVaR}_\alpha$ and $\mathbb{E}$ expressions in Equation 12.
>
> **How is adversarial welfare defined?**
>
> We can include a sentence in the preliminaries explaining that once an allocation is fixed, we call the minimum welfare over the uncertainty set the “adversarial” welfare.
>
> **Why is CVaR less important at lower noise levels?**
>
> At low noise levels, the CVaR measures (and optima) for sufficiently large values of $\alpha$ will be fairly close to the central estimates (and optima based on them) of welfare. As the noise increases, the variance increases. This results in long-tailed distributions, and thus the CVaR optimizer will produce different solutions from the optimizer of the central estimate. By definition, the robust approach is more sensitive than CVaR, which explains the observed differences between them. For further details on how CVaR provides robustness against high risks in fat-tailed distributions, please refer to Example 2.1 and [1].
>
> [1] R. Tyrrell Rockafellar and Stanislav Uryasev. (2002). Conditional Value-at-Risk for General Loss Distributions. Journal of Banking and Finance.
>
>
> **Can expand a bit on how these randomized rounding algorithms apply to your experiments?**
>
> Thank you for bringing up this point.
> Yes, indeed there is a generalization of Birkhoff-von-Neumann decomposition to multi-matchings [1]. [2] does a nice job of explaining its application in randomized reviewer assignments specifically.
>
> We should make this clearer in the paper, but we did not round the robust USW/GESW solutions reported in the experiments. In the case of the uncertainty-unaware USW/GESW and the sampling-based CVaR problem for both USW and GESW (the first 4 rows of all our tables), we were able to express these programs as MILP’s and directly solve for the optimal integer solution. We recommend using this approach whenever it is feasible. Theorem 14 of [3] demonstrates that although the rounded robust solutions may differ significantly from the fractional robust solutions, however, in the case of USW the welfare under the true valuations remains high after rounding. In our experiments, we do not assume access to ground truth valuations, so we cannot directly test this.
>
> Rounding often degrades the maximin objective value. In a simple analogy, imagine we have 2 coins. We have to pick one coin to flip, and we want it to be heads. An adversary can force one coin to always be tails. If we select a distribution over the coins, and sample from the distribution before the adversary selects their coin, we can obtain a non-zero probability of getting a heads. But it would be a very powerful adversary (perhaps unreasonably powerful) if they could see which coin we picked and force it to be tails.
> That being said, using the randomized rounding procedure with our methods results in allocations that are robust against adversaries in expectation.
>
> [1] Budish, E., Che, Y.K., Kojima, F., & Milgrom, P. (2009). Implementing random assignments: A generalization of the birkhoff-von neumann theorem.
>
> [2] Jecmen, S., Zhang, H., Liu, R., Shah, N., Conitzer, V., & Fang, F. (2020). Mitigating manipulation in peer review via randomized reviewer assignments. Advances in Neural Information Processing Systems, 33.
>
> [3] Cousins, C., Payan, J., & Zick, Y. (2023). Into the unknown: Assigning reviewers to papers with uncertain affinities. International Symposium on Algorithmic Game Theory, 16.

---

> > ### Comment · Reviewer_UQiJ · 2024-08-07
> >
> > The authors have addressed all my concerns and I have raised my score by one point. I found the submission very interesting, and think the paper will be a great contribution to the area!

---

> > > ### Author Response · Authors · 2024-08-07
> > > **Thank you!**
> > >
> > > Thank you so much for promptly reviewing our rebuttal and raising the score. We greatly appreciate it and are glad that you found our paper interesting!

---

### Official Review · Reviewer_69cR · 2024-07-10

**Soundness:** 3
**Presentation:** 3
**Contribution:** 3
**Rating:** 7
**Confidence:** 2

**Summary:**

This paper considers a new fair allocation problem. In the classical fair allocation problem, each item can only be matched with at most one agent, and the utilities are known prior. This paper considers a variant where each item is required to be matched with some number of agents. Agents can also be matched with some number of items. The utilities are not known prior. The problem is motivated by the paper-reviewer matching system. Each agent represents a reviewer and each item is paper. A paper is required to be assigned more than one reviewer. Each reviewer can be matched to more than one paper. In addition, each reviewer has a capacity that indicates the maximum number of papers assigned to her. The utilities are not known prior because the review quality is only known after the assignment.

The main contribution of this work is that they designed an efficient method to optimize both the utilitarian and egalitarian objectives.  The main technical tools come from the robust optimization area. In particular, they transform the problem into a pure linear programming problem via a series of propositions.

**Strengths:**

1. The paper is well-motivated. I like the model studied in this paper. The reviewer-paper matching algorithm is important. I expect that the proposed algorithmic idea should have the impact of a positive practice.

2. The paper is well-written. I am a theoretical person, I appreciate that the theory part of the paper is carefully organized. The whole proof idea is clear to me.

**Weaknesses:**

I have to say that I am usually working on approximation algorithms and I am not in the right position to judge the technical novelty of these continuous methods. To me, the downside is that the running time of the proposed algorithm is high, especially since the algorithm is required to solve LP. This limits the application of algorithms.

**Questions:**

I don't have any specific questions.

**Limitations:**

The societal impact is not available.

---

> ### Author Rebuttal · Authors · 2024-08-06
>
> Thank you for your comments!
>
> **To me, the downside is that the running time of the proposed algorithm is high, especially since the algorithm is required to solve LP. This limits the application of algorithms.**
>
>
> We acknowledge that some of our proposed algorithms have high runtimes. However, we would like to highlight the following points:
>
> 1. Optimizing fair welfare objectives under uncertainty is an NP-hard problem [3, 4]. Thus, obtaining exact solutions in polynomial time is not feasible.
>
> 2. Previous work in the literature on fair allocations and divisions has proposed methods such as MILPs [1, 2, 4] for solving fair allocation problems without considering uncertainty. Therefore, the runtimes of our solutions are comparable to existing methods.
>
> 3. Our iterated quadratic programming approach for robust measures with ellipsoidal uncertainty sets is significantly more efficient than the naive subgradient ascent method previously proposed, as shown in Figure 1.
>
> Finally, we recognize that our CVaR approach is limited by its sample complexity. For large instances of fair allocation problems, we recommend using the normal form of CVaR, which can be optimized using SOCP or Projected Gradient Ascent techniques.
>
> [1] Kawase, Y., Nishimura, K., Sumita, H. (2023). Fair Allocation with Binary Valuations for Mixed Divisible and Indivisible Goods. Arxiv.
>
> [2] Caragiannis, I., Kurokawa, D., Moulin, H., Procaccia, A., Shah, N., Wang, J. (2019). The Unreasonable Fairness of Maximum Nash Welfare. ACM Transactions on Economics and Computation, 7.
>
> [3] Cousins, C., Payan, J., & Zick, Y. (2023). Into the unknown: Assigning reviewers to papers with uncertain affinities. International Symposium on Algorithmic Game Theory, 16.
>
> [4] Peters, D., Procaccia, A.D., & Zhu, D. (2022). Robust Rent Division. Advances in Neural Information Processing Systems, 36.

---

> > ### Comment · Reviewer_69cR · 2024-08-11
> >
> > I'd like to thank the authors for replying to my concerns. I still think that the running time restricts the application of algorithms. But I believe that the studied problem is very interesting. Considering that I am not able to judge the technical novelty of this paper, I will keep my score and confidence unchanged.

---

> > > ### Author Response · Authors · 2024-08-14
> > > **Thank you!**
> > >
> > > Thank you again for your time and effort in reviewing our paper, and for both the positive comments and critiques.

---

### Official Review · Reviewer_5Awd · 2024-07-12

**Soundness:** 3
**Presentation:** 3
**Contribution:** 3
**Rating:** 7
**Confidence:** 4

**Summary:**

This paper looks at the problem of computing resource-agent matchings under uncertainty in a group setting, where the actual valuations of agent-resource matchings are unknown. When the distribution of the valuation uncertainty is known, they look at stochastic optimization using Conditional Value at Risk (CVaR), and when a set of candidate valuations is known, they look at robust optimization through max-min optimization for worst-case valuation. They present methods to optimize for two objectives: utilitarian, where each individual is equally weighted, and egalitarian, where they look at only the group with the minimum utility. They consider uncertainty sets which are either linear or elliptical, and provide derivations of the optimization algorithm for each combination, reducing the solution to an LP in many cases, and with quadratic programming or sub-gradient ascent in others. They also show some empirical results on a reviewer assignment dataset.

**Strengths:**

The paper formalizes a set of methods to optimize the important problem of fairness and group-aware resource allocation under uncertainty. The paper is well-written, and sufficient description is given for readers to comfortably follow the theory developed. The solutions to the optimization problems are reduced to computationally tractable instances that are popularly solvable, and the analysis for both linear and quadratic uncertainty sets is a welcome addition. The empirical results also provide context on how well certain solution approaches scale. The approach seems to be novel, and encompasses a range of general use cases.

**Weaknesses:**

1. One of the limitations I found in this paper was that the authors did not compare the uncertainty-aware solutions to other prior approaches at solving these problems: what benefit does this approach give us? There is no discussion regarding this. From the results, it seems the base methods (USW and GESW) are better than the Rob and CVaR variants.
2. The results do not show any indication of where the GESW optimization may be beneficial. Instead, it appears that the USW optimization supersedes the GESW optimization. If that is the case, the importance of the 'fairness' part of the paper is greatly diminished. The authors should preferably select an evaluation dataset where the approach can show its benefits, if any, to justify its inclusion.
3. As the authors state, the CVaR optimization is much slower. It also seems to require more information than the robust variant of the problem. Without a sufficient justification for its use, what is the merit of including it?

**Questions:**

1. What is the benefit of using the CVaR and Robust variations of the optimization when the base optimization performs better in most cases? Except adversarial welfare for the robust optimization, there seems to be no meaningful benefit of considering uncertainty.
1.1. Unless I am mistaken, the baseline USW and GESW methods are supposed to be improved by considering uncertainty. If this is not the case, this needs to be mentioned in the paper and clarified.
2. Why would we need to select the GESW objective when USW performs better in almost all cases? What is the significance of the fairness aspect of the paper in this case?
3. Where can this be applied? Can you give some examples of allocation problems that are admissible based on the assumptions and limitations of the approaches (e.g. linear/quadratic valuations), and cases where it would not hold?

**Limitations:**

The paper addresses some limitations in the text. The authors could expand on the situations in which the proposed methods offer an advantage. The authors should also mention the possible negative side effects of using methods that do not lead to intuitive solutions like using CVaR (in Example 2.1, for example). This may lead to affected users perceiving unfairness or inefficiency when the metric is not easy to understand.

---

> ### Author Rebuttal · Authors · 2024-08-06
>
> Thank you for your comments and suggestions! Please see our replies below.
>
> **Compare the uncertainty-aware solutions to other prior approaches at solving these problems?**
>
> We first emphasize that while we present novel solutions, our primary contribution is suggesting new *objectives* for the constrained allocation problem, i.e., optimizing utilitarian and egalitarian welfares under uncertainty. To our knowledge, the only existing prior work that solves one of these objectives for the constrained allocation problem is [1], which optimizes utilitarian welfare under ellipsoidal uncertainty sets using a projected subgradient ascent method. We compare against this method in the right half of Figure 1. Any algorithm that optimizes the uncertainty-unaware objectives is also an appropriate baseline. Those objectives can be optimized exactly, and we did compare against those uncertainty-unaware optimal baselines (USW and GESW in lines 1 and 2 in each table, and the dashed lines in the left half of Figure 1). We note that [5] focuses on optimizing expected envy or the probability of envy under uncertainty in valuations, which is a fundamentally different objective and thus not comparable to our methods. This distinction also applies to the other prior works mentioned in the related works section. We are happy to compare to other relevant baselines if we are made aware of them.
>
> Our global response on "Technical challenges in combining fairness and uncertainty" also applies to this question.
>
> **As the authors state, the CVaR optimization is much slower. It also seems to require more information than the robust variant of the problem. What is the merit of including it?**
>
> The Conditional Value at Risk (CVaR) of a random variable $ X$ at the $\alpha$ percentile is the expected value of $ X $ below its $\alpha$ quantile. By optimizing allocations to maximize the CVaR of welfare, we ensure that the welfare value will be at least $ w$ with $ 1-\alpha $ probability, where $w$ is the CVaR of welfare corresponding to the CVaR-optimal allocation. Put simply, instead of picking an allocation that ensures a welfare of $\ge w$ for *every* allocation (as does the max-min approach), CVaR allows a more refined analysis, allowing stakeholders to ensure a welfare of at least $w$ for at least (say) $90\%$ of the possible valuations.
>
> Thus, the CVaR measure is a less conservative objective than the Robust measure and is widely used in Operations Research and Machine Learning literature [2,3,4] to handle fat-tailed loss distributions.
>
> Finally, we emphasize that we do not advocate for any specific uncertainty-aware objectives; we simply demonstrate how to achieve fair allocations under uncertainty using popular uncertainty-aware objectives.
>
>
> **What is the benefit of using the CVaR and Robust variations of the optimization when the base optimization performs better in most cases?**
>
> The uncertainty-unaware USW and GESW optimization performs very poorly on the robust objectives (the last 2 columns of the tables). In Table 1, the CVaR of USW/GESW is optimal under the uncertainty-unaware baselines, but this is likely due to the fact that there is not very much variance in the distributions we estimate. This is evidenced by Figure 1, where performance of the uncertainty-unaware solutions drops as noise increases. However, we do expect the diagonal entries of all tables to be at, or very close to, 1.0 – i.e., when you (approximately) optimize for some objective, you perform near-optimally on that same objective.
>
> As shown on the left of Figure 1, the value of the uncertainty-unaware USW and GESW solutions on the CVaR objective becomes worse as the level of uncertainty increases. In our setting, the CVaR measure is particularly useful when the distribution of welfare is long-tailed, where optimizing for the worst case results in an overly conservative allocation (low welfare). We will use our extra page to add this clarifying discussion in the paper.
>
> **Why would we need to select the GESW objective when USW performs better in almost all cases? What is the significance of the fairness aspect of the paper in this case?**
>
> Please see the section of the global response on "Different ways of grouping reviewers".
>
> **Can you give some examples of allocation problems that are admissible based on the assumptions and limitations (e.g. linear/quadratic valuations) of the approaches?**
>
> We refer the reviewer to Appendices B, D, and E for detailed discussions on allocation problems where our methods are applicable.
>
> Please see the global response  and our response to Weakness 3 (W3) of reviewer hqEo for discussion on assumptions made in the paper.
>
> We also note that we have also included a discussion on the runtimes of our proposed algorithms in the global response.
>
> [1] Cousins, C., Payan, J., & Zick, Y. (2023). Into the unknown: Assigning reviewers to papers with uncertain affinities. International Symposium on Algorithmic Game Theory, 16.
>
> [2] Soma, T., Yoshida, Y. (2020) Statistical Learning with Conditional Value at Risk. Arxiv.
>
> [3] Rockafellar, R.T., and Uryasev, S. (2000). Optimization of Conditional Value-at-Risk. Journal of Risk, 2.
>
> [4] Stoyanov, S., Rachev, S., Racheva-Iotova, B., Fabozzi, F. (2011) Fat-tailed models for risk estimation. Journal of Portfolio Management, 37.
>
> [5] Dominik Peters, Ariel D. Procaccia, David Zhu (2022). Robust Rent Division. Advances in Neural Information Processing Systems.

---

> > ### Comment · Reviewer_5Awd · 2024-08-07
> >
> > Thanks for answering my questions and concerns.
> >
> > The new experiment included in the global response helps show the value of the GESW function. This should be included in the main paper.
> >
> > I am still not convinced about the need to include CVaR optimization. I agree that it allows better analysis, and it may be widely used, but what is the contribution it makes in the context of this paper? I understand that the paper's contribution is not in designing a better solution to an existing problem, but to define a new optimization. Yet, the benefits of using this optimization could be made clearer.
> >
> > From Figure 1, yes, the uncertainty-unaware solutions do become worse as the level of uncertainty increases, but so do the uncertainty-aware solutions (especially looking at USW, the difference is negligible). Can the authors comment on why this is, and perhaps give an example where considering uncertainty helps over the uncertainty-unaware approaches (specifically for CVaR)?

---

> ### Author Response · Authors · 2024-08-09
> **Use-cases of CVaR and the negligible difference between CVaR and USW in Figure 1**
>
> Thank you for asking these important questions.
>
> The utility of CVaR is higher when welfare distributions exhibit a left fat-tail, meaning a greater probability mass is concentrated in the left tail. Unlike the robust (minimax) approach, the CVaR method—particularly at higher $\alpha$ values—balances between extreme pessimism and optimizing for the average performance. This is advantageous in allocation problems with high uncertainty, where worst-case optimization can lead to overly pessimistic and inefficient outcomes.
>
> For example, in public housing assignments, where families have uncertain preferences for various housing units, worst-case optimization might result in inefficient allocations, leaving many families dissatisfied. By focusing on the worst $\alpha$-percentile preferences through CVaR, allocations can be made more efficiently (while also being robust to uncertainty with high probability), thereby improving overall satisfaction.
>
> However, it's crucial to differentiate between when to apply CVaR and when the robust approach is more appropriate. The robust approach is better suited to scenarios with extremely high stakes, where any failure (no matter how small the probability of occurrence) is unacceptable—such as life-or-death situations (e.g., allocating medical supplies after a disaster). It is also effective in low-uncertainty contexts where optimizing for the worst case is reasonable and doesn't significantly reduce efficiency.
>
> There are several reasons why CVaR USW behaves very similarly to the uncertainty-unaware USW maximal in Figure 1.
> 1) When the valuations are sampled from independent Gaussian distributions, the USW is just the mean of independent Gaussian variables. The variance of the utilitarian welfare is $\\frac{\\sum_a\\sum_i\\sigma_{a,i}^2}{(nm)^2}$ where $\sigma_{a,i}$ is the standard deviation of the valuation of item $i$ according to agent $a$. Due to this the variance of the utilitarian measure is fairly low, and USW is a more stable measure compared to GESW.
> 2) We were sampling valuations from symmetrical Gaussian distributions and so the noise in the valuations was (mostly) getting averaged out.
> 3) We also had a large number of items with very small variance. In AAMAS 2015 and 2016, around 8-9% of the entries have variance less than 0.005.
>
> We verified that when we model valuations using a negatively-skewed Gaussian distribution with the same means and variances [1], we see increasing robustness of CVaR relative to uncertainty-unaware USW. The difference is sharper as the skew parameter gets more negative. For example, on the AAMAS 2015 dataset we tried the following experiment where we sample valuations from a skewed-Gaussian distribution with varying skew parameter. We optimize and evaluate for CVaR$_{0.3}$.
> | Skew | CVaR USW |  USW |
> |:----:|:--------:|:----:|
> | -0.5 |   1.64   | 1.56 |
> |  -1  |   1.45   | 1.21 |
> |  -2  |   1.33   | 0.96 |
> |  -5  |   1.29   | 0.84 |
> |  -10 |   1.28   | 0.82 |
>
>
> [1] https://docs.scipy.org/doc/scipy/reference/generated/scipy.stats.skewnorm.html
>
> Note that in practice, if the valuation matrix is not Gaussian-distributed, we can generate samples from its posterior distribution using Markov Chain Monte Carlo (MCMC) or Variational Inference (VI).

---

> ### Author Response · Authors · 2024-08-09
> **Example where considering uncertainty (via CVaR approach) helps over the uncertainty-unaware approaches**
>
> The following toy example further demonstrates the robustness of CVaR when the welfare distribution has a fat-tail.
>
> Consider a scenario with two agents and four items, where each agent must be assigned one item. Each agent's valuation for an item is independent and follows a skewed Gaussian distribution. The mean valuations for the items are represented by the following 2D array:
> $$
> \\begin{bmatrix}
> 0.39 & 0.49 & 0.51 & 0.53 \\\\
> 0.52 & 0.51 & 0.53 & 0.54
> \\end{bmatrix}
> $$
>
> The standard deviations for the Gaussian distributions of the four items are $[0.01, 0.04, 0.05, 0.09]$, with a skewness factor of 5 across all distributions [1].
>
> We aim to optimize the utilitarian welfare, using CVaR$_{0.04}$ as our evaluation metric. This choice maximizes the expected utilitarian welfare over the worst $\alpha = 0.04$ quantile.
>
> To achieve this, we sampled 20,000 valuation matrices from the aforementioned distribution of valuations. We then applied three different optimization approaches: the CVaR approach at $\alpha = 0.04$, the Robust approach, and the Naïve approach that optimizes against the mean valuation.
>
> The following results were observed on a test set of 20,000 valuations sampled from the same distribution:
>
> **CVaR approach:**
>
> *Allocation:*
> $$
> \\begin{bmatrix}
> 0 & 0 & 1 & 0 \\\\
> 1 & 0 & 0 & 0
> \\end{bmatrix}
> $$
> *Test CVaR$_{0.04}$ utilitarian welfare:* 0.985
>
>
> **Robust approach:**
>
> *Allocation:*
> $$
> \\begin{bmatrix}
> 0 & 1 & 0 & 0 \\\\
> 1 & 0 & 0 & 0
> \\end{bmatrix}
> $$
> *Test CVaR$_{0.04}$ utilitarian welfare:* 0.972
>
> **Naïve approach:**
>
> *Allocation:*
> $$
> \\begin{bmatrix}
> 0 & 0 & 0 & 1 \\\\
> 0 & 0 & 1 & 0
> \\end{bmatrix}
> $$
> *Test CVaR$_{0.04}$ welfare:* 0.947
>
> **Observations**
>
> The naïve approach selects items 3 and 4, as they have the highest mean values, but it does not account for the uncertainty in the preferences. The robust approach, being more conservative, chooses items 1 and 2 due to their lower uncertainty.
>
> The CVaR approach strikes a balance between these two methods, selecting items 1 and 3. Item 1 has a lower mean value with low uncertainty, while item 3 has a higher mean value but with slightly higher uncertainty than item 2.
>
> We then repeat the experiment, this time optimizing for egalitarian welfare across agents, and evaluate the results by measuring CVaR$_{0.04}$ of the egalitarian welfare.
>
> **CVaR approach:**
>
> *Allocation:*
> $$
> \\begin{bmatrix}
> 0 & 0 & 1 & 0 \\\\
> 1 & 0 & 0 & 0
> \\end{bmatrix}
> $$
> *Test CVaR$_{0.04}$ egalitarian welfare:* 0.47
>
>
> **Robust approach:**
>
> *Allocation:*
> $$
> \\begin{bmatrix}
> 1 & 0 & 0 & 0 \\\\
> 0 & 1 & 0 & 0
> \\end{bmatrix}
> $$
> *Test CVaR$_{0.04}$ egalitarian welfare:* 0.38
>
> **Naïve approach:**
>
> *Allocation:*
> $$
> \\begin{bmatrix}
> 0 & 0 & 0 & 1 \\\\
> 0 & 0 & 1 & 0
> \\end{bmatrix}
> $$
> *Test CVaR$_{0.04}$ egalitarian welfare:* 0.45
>
> **Observations**
>
> We notice a significant decline in the performance of the robust approach. This decline occurs because, although items 1 and 2 have lower uncertainty compared to items 3 and 4, item 1 has significantly lower uncertainty than item 2, resulting in better worst-case utility for item 1. Since the CVaR approach selected items 3 and 1, it achieves a higher CVaR of egalitarian welfare at $\alpha = 0.04$ compared to the robust approach which selects items 1 and 2.
>
> We also see fairly similar results when setting $\alpha=0.2$ with the other problem parameters remaining the same.
>
> [1] https://docs.scipy.org/doc/scipy/reference/generated/scipy.stats.skewnorm.html

---

> > ### Comment · Reviewer_5Awd · 2024-08-12
> >
> > Thank you for responding to my follow-up. The example is useful in helping me understand when uncertainty-aware solutions are better. I would recommend the authors to add an experiment that demonstrates it, to make a stronger case for their paper.
> >
> > Considering our discussion during this period, I will increase my score by one point.

---

> > > ### Author Response · Authors · 2024-08-13
> > > **Thank you!**
> > >
> > > Thank you so much for promptly checking our response and raising the score! We sincerely appreciate your feedback and constructive criticism on our paper. We will incorporate the suggested changes in the camera-ready version.

---

### Official Review · Reviewer_ZdA8 · 2024-07-13

**Soundness:** 3
**Presentation:** 2
**Contribution:** 2
**Rating:** 6
**Confidence:** 1

**Summary:**

The authors study a resource allocation problem where the objective is to optimize for efficiency and fairness under the presence of some uncertainty. The agents are partitioned into groups and the items need to be assigned so as to be fair to the groups. They study two maximization objectives : 1) a weighted sum of social welfare of groups and 2) the social welfare of the group with the least amount of welfare. They study two models of uncertainty over the agents' valuations of the items - the first is when there is some explicit uncertainty set and the objective is to maximize the worst-case outcome across this uncertainty set, and the second is when the distribution over the valuations is known and the objective is to maximize the Conditional Value at Risk (CVaR$_\alpha$).

Their primary contribution is to present formulations of these problems as various mathematical programs. They make assumptions on the uncertainty sets so as to make these formulations suitable for optimization in certain cases. They present experiments supporting their models and formulations.

**Strengths:**

1) Their experiments suggest that optimizing for CVaR has some benefit under a large uncertainty regime, as well as points to the benefit of considering the robust uncertainty model.
2) Their formulation and subsequent solution appears to be faster, and more successful, than applying the naive solver.

**Weaknesses:**

The paper is a bit terse, and hard to follow at times. It would be beneficial to add more details where necessary. For example, in Line 180, please mention the randomized rounding procedure or at least the properties of the output rounded solution. Ideally, there would at least be some more details to get to the propositions.

**Questions:**

1) Are there any tractable settings where the agents can be in multiple groups at once? There are many resource allocation, clustering, etc. papers where such settings are considered.
2) Can't the robust setting be viewed as a subcase of the CVaR setting where you maximize the worst (ie. the 0 percentile) outcome?

---

> ### Author Rebuttal · Authors · 2024-08-06
>
> Thank you for your comments and suggestions!
>
> **Please mention the randomized rounding procedure or at least the properties of the output rounded solution.**
>
> Thank you for the suggestion. We will add more details on the rounding procedure in the camera ready version of the paper. Please see the final answer in our response to Reviewer UQiJ as well, where we discuss the rounding procedure and its relation to our experiments in a bit more detail.
>
> In general, we can use the extra page in the camera-ready to expand some details that were necessarily compressed for the submission. Reviewer UQiJ also had a great suggestion to move some of the formal theorem statements to the appendix, which should free up additional space for explanations.
>
> **Are there any tractable settings where the agents can be in multiple groups at once?**
>
> Our LP solutions for robust allocation with linear uncertainty sets and CVaR allocation (without normality assumption) for both utilitarian and egalitarian welfare can be trivially extended to the case where agents are in multiple groups.
>
> This is a good point in the case of robust GESW. However, the independence of groups assumption is not a fundamental limit, but rather a simplifying assumption. The subgradient ascent approach still works for the robust GESW allocation when groups aren’t disjoint. Because the minimizer will concentrate the uncertainty on a single group for GESW anyway, when we compute the minimizer at each step of subgradient ascent, we can just compute the minimizer for each group individually (even though the groups have some overlap).
>
> We agree the disjoint groups assumption can be limiting, but not unreasonable, in practical scenarios. Conferences often group papers into disjoint tracks and/or require paper authors to submit a unique primary subject area. Although they may have multiple secondary subject areas, the top-level grouping remains independent. We can handle grouping by secondary subject area using subgradient ascent as mentioned above, but it is also reasonable to use the primary subject area grouping and apply the faster approach we describe in our paper.
>
> We’ll include all this discussion in the paper.
>
> **Robust setting be viewed as a subcase of the CVaR setting.**
>
> Yes, it is well-known in the literature that the robust setting is a special case of CVaR where $\alpha=0$. However, using the CVaR approach proposed in Section 3 is inefficient because it requires a large number of samples to ensure that the uncertainty set implicitly constructed by CVaR captures the worst-case valuation matrix. Instead, we can directly incorporate the learned uncertainty set as a constraint and optimize against the worst-case model in this uncertainty set by solving a max-min optimization problem. This results in a more efficient algorithm that is not limited by the sampling complexity of CVaR.
>
> We note that prior works [1,2,3] in Distributional Robust Optimization have taken similar approaches when dealing with worst-case objectives.
>
>
> [1] Rahimian, H., Mehrotra, S. (2019). Distributionally Robust Optimization Review. Arxiv.
>
> [2] Lobo, E., Ghavamzadeh, M., Petrik, M. (2020). Soft-Robust Algorithms for Batch Reinforcement Learning. Arxiv.
>
> [3] Virginie Gabrel, Cécile Murat, Aurélie Thiele. (2014). Recent Advances in Robust Optimization: An Overview. European Journal of Operations Research

---

### Official Review · Reviewer_hqEo · 2024-07-15

**Soundness:** 3
**Presentation:** 2
**Contribution:** 2
**Rating:** 4
**Confidence:** 2

**Summary:**

The authors of this paper investigate the fair multi-matching problem under uncertainty. Both stochastic and robust optimizations are considered to solve the proposed problem.

**Strengths:**

S1. Fairness is an important and practical concern in resource allocation problems.

S2. The theoretical results of this paper seem to be correct.

**Weaknesses:**

W1. The authors fail to provide a clear motivation example of the proposed research problem in real applications. For the reviewer assignment application, I do not see strong reasons why we need to consider uncertainty (the reviewers have already revealed their preferences over papers clearly) and fairness (the utility of a reviewer is how the assigned papers match her/him?).

W2. What are the major technical challenges of considering both group fairness and uncertainty in the resource allocation problem? Since there are works discussing each factor (fairness and uncertainty), can we just adapt existing techniques to solve the proposed problem? Are there any new technical challenges caused by the combination of fairness and uncertainty?

W3. Does the process of obtaining the uncertainty set assume a true model (usually linear models in statistics) of the uncertain parameters? If so, I am afraid the way of considering uncertainty is still a toy research problem because in reality the data distribution is usually unknown and complicated. The uncertainty set itself may not be constructed in a reliable way. For example, under what conditions can the authors prove that their uncertainty sets described in Appendix D have the true parameters? To prove this, I guess the authors may need strong assumptions about the data distribution.

W4. As the reviewer assignment datasets do not have groups of reviewers, the authors may want to try different ways of grouping reviewers in experiments to make the results more convincing w.r.t. group fairness.

W5. It seems to me that the experiments are just numerical simulations where the objectives are simulated. This makes the research problem a toy one as no real datasets are used to verify the proposed method.

**Questions:**

W1, W2, W3

**Limitations:**

W1, W3, W4, W5

---

> ### Author Rebuttal · Authors · 2024-08-06
>
> Thank you for your detailed review. We respond to your comments and questions below.
>
> ## W1
>
> As mentioned on line 306, we adopt the model used by several major conferences: ICML 2022, AAAI 2022-2024, and IJCAI 2022-2024 [1]. In this model, papers are the agents, reviewers are the items, and the value $V_{a,i}$ of assigning reviewer $i$ to paper $a$ is estimated from multiple sources to predict the overall value to the conference of eliciting that review.
>
> As such, the reviewer assignment problem offers several sources of uncertainty. Reviewers offer extremely sparse partial rankings of papers, with a significant proportion of submissions remaining unranked by *any* reviewer [1, 3]. Affinity score computation systems (e.g. those generated by TPMS [4] and Open Review’s expertise model [5]) utilize NLP methods which have
> well-documented error rates [5]. [6, 7] both give strong overviews of the components that go into modern automated reviewer assignment systems, along with additional discussion of the noise inherent in the process. We can include all of this discussion in the camera ready.
>
> ## W2
>
> Please see the global response.
>
> ## W3
>
> Modeling uncertainty and constructing uncertainty sets are not contributions of this paper. Our main contribution is to show that for certain commonly used uncertainty sets (see, e.g., [2] for derivations), the problem of optimizing fair welfare objectives under uncertainty can be efficiently solved using our proposed methods. Cousins et al. [7] also provide more detail on constructing uncertainty sets in the same setting as the current work.
>
> That being said, Appendix D illustrates a simple example of how to construct an uncertainty set. The simplified bound in Appendix D assumes 1) that the validation set is drawn from the same distribution as the distribution under which we make assignments, and 2) that the cross-entropy loss is normally distributed. The first assumption, although quite strong, is fairly standard in machine learning (see Ch. 6 and 7 of Shalev-Shwartz and Ben-David [8] or Ch. 14 of Mitzenmacher and Upfal [9]). There are also ways to relax this assumption by modeling distribution shift or bounding the total variation distance between the distributions. The second assumption is fairly standard as well, as it holds in the limit by the central limit theorem. While there are certainly more accurate ways of estimating this generalization error, these improvements would only change the real-valued RHS of the inequality between lines 574-575 or add fixed multiplier terms on the LHS (more details in [7]). In both cases the structure of the optimization problem (our main focus in this work) remains unchanged.
>
> ## W4
>
> Please see the global response.
>
> ## W5
>
> We do try to generate results using settings that are as realistic as possible (e.g. basing our experiments on a mix of empirical data and simulated data), however they are ultimately not ‘the real deal’. We have spent a considerable amount of time and effort unsuccessfully trying to convince conference organizers to provide us with some access to reviewer data,
> with appropriate anonymity and experimental standards in place. Organizers’ reluctance to support meta-analysis of reviewer assignment is (to some extent) understandable: the stakes of conference reviewing are extremely high, and the organizers bear some overhead in supporting the experiments we want to run on real data.
>
> We also suffer from a chicken-and-egg problem: in order to make the case to conference organizers that our methods make sense, they need to be published in top-tier venues. In order to get our methods published in top-tier venues, we need access to data by conference organizers.
>
> ### References
>
> [1] Leyton-Brown, K., Mausam, Nandwani, Y., Zarkoob, H., Cameron, C.,
> Newman, N., & Raghu, D. (2024). Matching Papers and Reviewers at Large
> Conferences. Artificial Intelligence, 331.
>
> [2] Gupta, V. (2019). Near-Optimal Bayesian Ambiguity Sets for Distributionally Robust Optimization. Management Science, 65.
>
> [3] Rozenzweig, I., Meir, R., Mattei, N., & Amir, O., (2023). Mitigating Skewed
> Bidding for Conference Paper Assignment. International Conference on Autonomous Agents and Multiagent Systems, 22.
>
> [4] https://torontopapermatching.org/webapp/profileBrowser/about_us/
>
> [5] https://github.com/openreview/openreview-expertise?tab=readme-ov-file#performance
>
> [6] Shah, N. (2022). Challenges, experiments, and computational
> solutions in peer review. Communications of the ACM, 65.
>
> [7] Cousins, C., Payan, J., & Zick, Y. (2023). Into the unknown: Assigning reviewers to papers with uncertain affinities. International Symposium on
> Algorithmic Game Theory, 16.
>
> [8] Shalev-Shwartz, S. and Ben-David, S. (2014). Understanding Machine Learning: From Theory to Algorithms.
>
> [9] Mitzenmacher, M. & Upfal, E. (2017) Probability and Computing, 2nd edition.

---

### Author Rebuttal · Authors · 2024-08-06

Thank you to all the reviewers for your detailed and thought-provoking reviews. We have responded to most of your points individually, but a few points were worth addressing globally.

**Algorithm runtime/scaling**

We acknowledge that some of our proposed algorithms have high runtimes. However, we would like to highlight the following points:

1. Optimizing fair welfare objectives under uncertainty is an NP-hard problem [3, 4]. Thus, obtaining exact solutions in polynomial time is not feasible.

2. Previous work in the literature on fair division has proposed methods such as MILPs [2, 4, 5] for solving fair allocation problems without considering uncertainty. Therefore, the runtimes of our solutions are comparable to existing methods.

3. Our iterated quadratic programming approach for robust measures with ellipsoidal uncertainty sets is significantly more efficient than the naive subgradient ascent method previously proposed, as shown in Figure 1.

Finally, we recognize that our CVaR approach is limited by its sample complexity. For large instances of fair allocation problems, we recommend using the normal form of CVaR, which can be optimized using SOCP or Projected Gradient Ascent techniques.

**Technical challenges in combining fairness and uncertainty**

Satisfying fairness notions under deterministic valuations, and optimizing under uncertainty are both rich problems in their own right; combining the two makes the problem that much more difficult. Optimizing the egalitarian welfare objective without uncertainty is already an NP-hard ILP [1]. Robust optimization of a linear objective with integer decision variables is already NP-hard in general as well [3]. When we combine the two, we have a constrained integer max-min problem that is difficult to solve for general uncertainty sets. We therefore must determine how the approaches common in the robust and stochastic optimization literatures adapt to the new objective, and if any simplified forms admit more efficient solutions. The main contribution of this paper is to identify instances of that problem that can be solved exactly or approximated efficiently. We demonstrate that for ellipsoidal and linear uncertainty sets, as proposed in distributionally robust literature, our problem can be simplified to more manageable forms. These can be efficiently solved using linear programming, iterated quadratic programming, or projected gradient ascent methods, as shown in Table 1.

**Different ways of grouping reviewers**

In the reviewer assignment problem, papers are the agents, and reviewers are the items. Therefore, we group papers rather than reviewers. Many conferences categorize papers based on their field or subfield of research. Ideally, each group of papers should receive a sufficient number of qualified reviewers, which we ensure via our proposed algorithms.

We have implemented a simulated example where there are 2 groups and 1 group is disadvantaged compared to the other. We took the AAMAS 2015 dataset, set the original papers to be group 1, and created a second group of papers by randomly selecting $k$ of the papers. For these $k$ papers, we divided the copied valuations by a number $d>1$, and set to $0$ all but the top $b$ valuations per paper. The defaults were $k=150, b=5$, and $d=2$; we tried varying each of these keeping the other two fixed. For each setting we compute the % of relative loss in GESW incurred by the max USW solution, or $\frac{f-g}{f}$, where $f$ is the GESW of the max GESW solution and $g$ is the GESW of the max USW solution. We show in our rebuttal PDF that under this setting, optimizing for USW instead of GESW results in sharp decreases in GESW, and the difference gets sharper as $k$, $b$, or $d$ increase.

**Assumption: Groups have independent uncertainty sets**

This assumption is not a fundamental limit, but rather a simplifying assumption for some cases. Our LP solutions for robust allocation with linear uncertainty sets and CVaR allocation (without normality assumption) for both utilitarian and egalitarian welfare can be trivially extended to the case where agents are in multiple groups. The subgradient ascent approach still works for the robust GESW allocation when groups are not independent. Because the minimizer will concentrate the uncertainty on a single group for GESW anyway, when we compute the minimizer at each step of subgradient ascent, we can just compute the minimizer for each group individually.

We agree the assumption can be limiting, but not unreasonable, in practical scenarios. Conferences often group papers into disjoint tracks and/or require paper authors to submit a unique primary subject area. Although they may have multiple secondary subject areas, the top-level grouping remains independent.

[1] Garg, N., Kavitha, T., Kumar, A. Mehlhorn, K., & Mestre, J. (2010). Assigning papers to referees. Algorithmica, 58.

[2] Caragiannis, I., Kurokawa, D., Moulin, H., Procaccia, A., Shah, N., Wang, J. (2019). The Unreasonable Fairness of Maximum Nash Welfare. ACM Transactions on Economics and Computation, 7.

[3] Cousins, C., Payan, J., & Zick, Y. (2023). Into the unknown: Assigning reviewers to papers with uncertain affinities. International Symposium on Algorithmic Game Theory, 16.

[4] Peters, D., Procaccia, A.D., & Zhu, D. (2022). Robust Rent Division. Advances in Neural Information Processing Systems, 36.

[5] Kawase, Y., Nishimura, K., Sumita, H. (2023). Fair Allocation with Binary Valuations for Mixed Divisible and Indivisible Goods. Arxiv.

---

### Decision · Program_Chairs · 2024-09-25

**Decision:**

Accept (poster)

**Comment:**

The reviews for this paper are largely positive, and I believe the paper should be accepted. There is one negative review, that seems to be sufficiently addressed by the authors; further, one of the reviewers has also highlighted in the discussion that some of the weaknesses pointed out by Reviewer hqEo may be relatively minor.